# Hierarchical Optimization via LLM-Guided Objective Evolution for Mobility-on-Demand Systems

**Yi Zhang**[1]* **Yushen Long**[2] **Yun Ni**[3] **Liping Huang**[1] **Xiaohong Wang**[1] **Jun Liu**[4]

[1] Agency for Science, Technology and Research, Singapore
[2] Morgan Stanley Asia Pte.
[3] Onto Innovation Inc.
[4] School of computing and communications, Lancaster University, UK
{Zhang_Yi, Huang_Liping, Wang_Xiaohong}@a-star.edu.sg
Yushen.Long@morganstanley.com, Yun.Ni@ontoinnovation.com
j.liu81@lancaster.ac.uk

## Abstract

Online ride-hailing platforms aim to deliver efficient mobility-on-demand services, often facing challenges in balancing dynamic and spatially heterogeneous supply and demand. Existing methods typically fall into two categories: reinforcement learning (RL) approaches, which suffer from data inefficiency, oversimplified modeling of real-world dynamics, and difficulty enforcing operational constraints; or decomposed online optimization methods, which rely on manually designed high-level objectives that lack awareness of low-level routing dynamics. To address this issue, we propose a novel hybrid framework that integrates large language model (LLM) with mathematical optimization in a dynamic hierarchical system: (1) it is training-free, removing the need for large-scale interaction data as in RL, and (2) it leverages LLM to bridge cognitive limitations caused by problem decomposition by adaptively generating high-level objectives. Within this framework, LLM serves as a meta-optimizer, producing semantic heuristics that guide a low-level optimizer responsible for constraint enforcement and real-time decision execution. These heuristics are refined through a closed-loop evolutionary process, driven by harmony search, which iteratively adapts the LLM prompts based on feasibility and performance feedback from the optimization layer. Extensive experiments based on scenarios derived from both the New York and Chicago taxi datasets demonstrate the effectiveness of our approach, achieving an average improvement of 16% compared to state-of-the-art baselines.

## 1 Introduction

Mobility-on-demand platforms, such as ride-hailing services, have become critical urban transportation infrastructures, to address unbalanced demand and supply by continuously executing two core decision-making processes: order dispatch and vehicle routing [1, 2, 3, 4, 5, 6]. This sequential decision-making involves solving complex combinatorial optimization problems under spatiotemporal constraints, ensuring timely and efficient service delivery in ever-evolving urban environments.

Firstly, reinforcement learning methods [7, 5, 8, 9, 10, 11] have been employed to learn policies to address the problem. These approaches can capture long-term rewards by considering the future trajectories of drivers and passengers. However, RL methods often require extensive training data and interactions to learn effectively. Training can be unstable and enforcing hard constraints is challenging. In addition, learned policies can behave unpredictably outside of their training distribution.

---

*Corresponding Author

39th Conference on Neural Information Processing Systems (NeurIPS 2025).

Secondly, two-stage optimization frameworks have been deployed to decompose the problem to reduce the complexity. Typically, the high-level stage addresses supply-demand balancing, e.g., batch matching via mixed-integer linear programming (MILP) [2, 3, 12], while the low-level stage focuses on dispatching or routing, e.g., graph-based search [13, 14]. However, high-level objectives are often manually designed without full knowledge of the low-level dynamics, potentially leading to suboptimal decisions. This disconnect can lead to misaligned objectives that do not fully reflect the true operational constraints, ultimately resulting in inefficiencies like increased waiting times or idle distances. Thirdly, large language models offer a new paradigm: their embedded priors on combinatorial reasoning and urban mobility can improve adaptability. However, current applications focus primarily on static optimization settings [15, 16, 17, 18, 19]. When applied to dynamic ride-hailing systems with real-time state transitions, these methods face two key limitations: (1) absence of iterative refinement aligned with evolving system states, and (2) lack of solver integration, resulting in proposals that may violate constraints or degrade solution quality. To address the above issue, we present the first integration of LLM and mathematical optimization for dynamic sequential decision-making systems. Specifically, we propose a hybrid LLM-optimizer framework that decomposes the problem hierarchically, strategically embedding LLM only where human expertise bottlenecks exist: (1) **LLM as Meta-Objective Designer**: Dynamically evolves high-level objectives via prompt-based harmony search [20, 21], guided by feasibility feedback from the optimization solver. (2) **Optimizer as Constraint Enforcer**: Solves low-level routing with mathematical rigor, ensuring real-time feasibility. (3) **Heuristics as Prompt Evolver**: Leverages harmony search algorithm to iteratively refine LLM prompts, guided by optimizer feedback to adaptively explore and converge toward effective meta-objectives. This framework is training-free, eliminating the need for extensive data or interaction required by RL-based methods. Simultaneously, by leveraging LLM to adaptively evolve the high-level objective, it mitigates the sub-optimality introduced by manual design in decomposed optimization, aligning high-level decisions more closely with downstream dynamics.

Our framework hierarchically decomposes each decision step into two levels: The high-level module is responsible for assigning passengers to taxis based on real-time spatial configurations and anticipated supply-demand imbalances, while the low-level module solves the routing or visiting sequence problem for each taxi to minimize passenger waiting time under spatiotemporal constraints. To address the partial observability challenge (high-level model lack foresight into downstream routing dynamics) induced by the decomposition, we employ LLM as a meta-heuristic designer, leveraging its implicit understanding of urban mobility patterns to adaptively refine high-level objectives. As captured in Figure 1, LLM generates high-level assignment objectives that serve as semantic guides within the optimization loop. These objectives are embedded into a closed-loop evolutionary process, where each simulation epoch evaluates their fitness. The evolutionary mechanism is guided by a harmony search algorithm, which iteratively refines the LLM prompt space to improve objective quality. This feedback-driven mechanism enables the LLM-generated heuristics to adapt and improve over time, combining the semantic richness of LLM with the structural robustness of traditional optimization. By integrating LLM as a semantic objective generator within a hierarchical optimization loop, our framework achieves dynamic adaptability absent in static operation research (OR) formulations, meanwhile avoiding the data inefficiency of RL-based approaches.

We summarize our contributions as follows: 1) We propose a hybrid LLM + Optimizer paradigm, where LLM acts as a meta-optimizer for evolving high-level objective through prompt-based evolution, while mathematical optimization solver guarantees constraint satisfaction and numerical rigor. To the best of our knowledge, this is the first work to leverage the capabilities of LLM combined with optimization solver for sequential dynamic decision-making systems. 2) We introduce a harmony search algorithm with three novel operators (random inference, heuristic improvement, and innovative generation) to iteratively refine LLM-generated objectives using feedback from the mathematical solver. 3) Extensive experiments on various scenarios derived from the New York and Chicago taxi datasets demonstrate the effectiveness of our approach, reducing passenger waiting time by approximately 16% over state-of-the-art baselines.

## 2 Related work

**Reinforcement Learning in Ride-Hailing** RL-based approaches have been adopted, which enables the system to learn from historical data and optimize decision-making by considering the expected

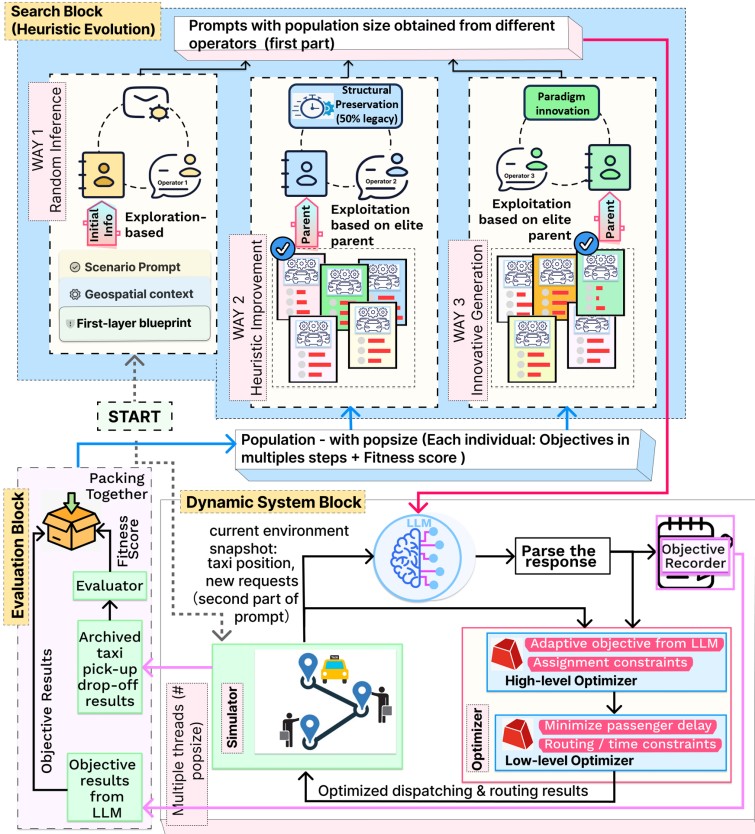

Figure 1: Overall control flow framework. **Search block**: Uses harmony search algorithm to iteratively select and apply 3 prompt-refinement operators. Initial iterations prioritize heuristics from Operator 1; **Dynamic System block**: At each timestep, the LLM generates high-level objectives based on the refined prompt and simulator-reported states (e.g., driver locations, pending orders). These objectives guide a two-level optimizer: a high-level dispatcher assigns orders, and a low-level router determines feasible visiting sequences. The optimizer's decisions are executed in the simulator, updating system states. This closed-loop process continues until the simulation horizon concludes; **Evaluation block**: Computes the fitness score from simulator trajectories and pairs it with the LLM-inferred objectives to update the harmony search population.

future trajectories of drivers and passengers [7, 10, 4, 22], including the deep RL-based approaches [11, 5, 8, 9, 23]. [6] uses RL to estimate the long-term value of each possible driver-order assignment, considering not just immediate trip rewards but also future opportunities (like driver repositioning). [10] integrates behavior prediction and combinatorial optimization with a deep double scalable network to generate order-driver assignments in an auto-regressive manner. However, RL-based approaches often require a large number of interactions to learn effectively, and their training can be unstable and highly sensitive to hyperparameters.

**On-line Optimization in Order Dispatching** Order-taxi matching problems in ride-hailing platforms are typically formulated as MILP models or framed as bipartite graph matching problems [2, 3, 12, 13, 14, 24]. Combinatorial optimization problems often face scalability challenges due to the large decision variable space, leading to multi-stage decompositions. [2] proposed a federated optimization framework integrating assignment blocks and routing engines with limited information exchange. [24] developed two MILP models for ride-hailing: a flow-based high-level model for supply-demand balance and a routing-based low-level model to minimize travel time. However, hierarchical decomposition introduces sub-optimality by decoupling optimization stages: high-level objectives lack visibility into low-level operational dynamics, resulting in myopic decisions that fail to preserve system-wide optimality.

**LLM for Operation Research Problems** Recent studies have explored the use of LLMs to automate the solution of OR problems, with approaches falling into two main categories: (1) automating mathematical model formulation to reduce reliance on domain expertise [18, 17, 25, 26, 19, 27]. Approaches such as task allocation, few-shot learning, and chain-of-experts have been proposed to guide programmatic LLM prompts [28, 29, 30, 17, 31, 32]. (2) Directly querying LLMs for algorithmic code to find feasible solutions to OR problems [15, 33, 34, 35, 36]. FunSearch [15] leverages the generative capabilities of LLMs to propose candidate programs, which are then rigorously evaluated to identify high-quality solutions. The EoH framework [16] encodes heuristic concepts as natural language "thoughts", which are then translated into executable code by LLM. Additionally, it employs an evolutionary algorithm to iteratively refine and evolve the prompts. However, to the best of our knowledge, no studies have yet combined LLM with optimization formulations in the context of dynamic decision-making systems.

## 3   Methodology

### 3.1   Dynamic Hierarchical Optimization Problem

We propose a novel problem formulation for ride-hailing system (Appendix A.6) with a formal analysis of the associated search space complexity (Appendix A.5), which jointly motivate the decomposition of the original task into two tractable subproblems - involving sequential decisions in task assignment and vehicle routing. These subproblems are integrated within a hierarchical framework augmented by an LLM component.

**First-level assignment problem** As an assignment problem, our goal extends beyond merely determining the next immediate passenger for an idle taxi. Instead, we formulate a comprehensive assignment across an entire system snapshot, ensuring that each passenger is served and assigned to at most one taxi. Omitting additional variables that capture taxi dynamics at the first level significantly reduces problem complexity. However, traditional approaches employ handcrafted objectives (left below), but these lack awareness of low-level routing dynamics may lead to suboptimal system performance:

$$
\begin{aligned}
\min \quad & J_{1st} \\
\text{s.t.} \quad & \sum_v y^{pv} = 1 \\
& y^{pv} \in \{0,1\}
\end{aligned}
\qquad
\begin{cases}
J_{1st}^{\text{dist}} = \sum_{p,v} y^{pv}\left(TR_{O^PS^v} + TR_{D^PS^v}\right) & \text{(Distance)} \\
J_{1st}^{\text{time}} = \sum_{p,v} y^{pv}\left|T^p - t_{S^v}\right| & \text{(Temporal)} \\
J_{1st}^{\text{util}} = \sum_v \left(\sum_p y^{pv}\right)^2 & \text{(Utilization)}
\end{cases}
\xRightarrow{\text{LLM Refine}}
\begin{aligned}
\min \quad & \Phi_t = \text{LLM}(\mathcal{S}_t, H_{t-1}) \\
\text{s.t.} \quad & \sum_v y^{pv} = 1 \\
& y^{pv} \in \{0,1\}
\end{aligned}
$$

**Notes:** $O^p, D^p$: Passenger $p$ origin, destination points; $S^v$: Taxi $v$ start position. $TR_{ij}$: Travel time $i \to j$; $T^p$: Passenger request time; $t_{S^v}$: Taxi $v$ start time. $y^{pv} \in \{0,1\}$: Binary assignment variable.

These objective components can be linearly combined through weighted summation: $J_{1st} = \alpha J_{1st}^{\text{dist}} + \beta J_{1st}^{\text{time}} + \gamma J_{1st}^{\text{util}}$, where fixed weights $\alpha, \beta, \gamma$ trade off distance (vehicle km), temporal alignment (estimated waiting time) and utilization fairness. While this myopic approach lacks visibility into low-level dynamics, our framework replaces handcrafted objectives with LLM-generated $\Phi_t = \text{LLM}(\mathcal{S}_t, H_{t-1})$, where $\mathcal{S}_t$ encodes vehicle positions and pending requests, and $H_{t-1}$ captures historical congestion and assignments. By embedding latent urban dynamics into MILP objectives, the LLM bridges the gap between assignment and routing without explicit dynamic modeling.

**Second-level sequencing problem** After solving the first-level problem and assigning passengers to each taxi, the second-level problem can be solved independently for each taxi, eliminating index $v$. In the formulation below, $O_p$ and $D_p$ represent the origin and destination of passenger $p$, while $S$ and $E$ denote the taxi's source and sink depots. $\tilde{t}$ is the estimated arrival time. The pickup service point set $\mathcal{P}$ consists of $\{(p, O^p)\}$, and the dropoff service point set $\mathcal{D}$ consists of $\{(p, D^p)\}$.

Constraints (1b)-(1d) initialize location and arrival time assignments, where $x_{ij}$ is a binary variable indicating link selection. $AR_i$ and $DP_i$ denote arrival time and depart time at position $i$. Constraints (1e)-(1f) enforce flow conservation, ensuring equal incoming and outgoing links. Constraints (1g)-(1m) define system dynamics: (1g)-(1h) establish taxi arrival times at dropoff points as the sum of pickup departure times and travel durations, while (1j)-(1k) enforce the same for new pickups.

Departure times must not precede arrival times (1i), (1l), and must larger than or equal to passenger request times (1m). The objective (1a) minimizes total passenger waiting time.

$$\text{minimize} \quad J_{2nd} = \sum_p (DP_{(p,O^p)} - T^p) \tag{1a}$$

$$\sum_{i \in \mathcal{P}} x_{S,i} = 1 \tag{1b}$$

$$AR_S = \tilde{t} \tag{1c}$$

$$\sum_{i \in \mathcal{P}} x_{i,E} = 1 \tag{1d}$$

$$\sum_{i \in \mathcal{P}} x_{ij} = \sum_{q \in \mathcal{P}} x_{jq}, \forall j \in \mathcal{D} \tag{1e}$$

$$\sum_{j \in \mathcal{D}} x_{ji} = \sum_{k \in \mathcal{D}} x_{ik}, \forall i \in \mathcal{P} \tag{1f}$$

$$AR_{(p,D^p)} \leq DP_{(p,O^p)} + TR_{O^p D^p} + M(1 - x_{(p,O^p),(p,D^p)}) \tag{1g}$$

$$AR_{(p,D^p)} \geq DP_{(p,O^p)} + TR_{O^p D^p} - M(1 - x_{(p,O^p),(p,D^p)}) \tag{1h}$$

$$DP_{(p,D^p)} \geq AR_{(p,D^p)} \tag{1i}$$

$$AR_{(p',O^{p'})} \leq DP_{(p,D^p)} + TR_{D^p O^{p'}} + M(1 - x_{(p,D^p),(p',O^{p'})}) \tag{1j}$$

$$AR_{(p',O^{p'})} \geq DP_{(p,D^p)} + TR_{D^p O^{p'}} - M(1 - x_{(p,D^p),(p',O^{p'})}) \tag{1k}$$

$$DP_{(p',O^{p'})} \geq AR_{(p',O^{p'})} \tag{1l}$$

$$DP_{(p',O^{p'})} \geq T^{p'} \tag{1m}$$

## 3.2 LLM-Optimizer Interaction Protocol

### 3.2.1 Scenario prompt setup

The scenario prompt $\mathcal{P}_{sce}$ equips the LLM with contextual knowledge regarding the problem, the assigned role, and the specific task. The prompt structure is formalized through three nested layers: $\mathcal{P}_{sce} = \mathcal{P}_{sys} \oplus \mathcal{P}_{geo} \oplus \mathcal{P}_{model}$. (1) $\mathcal{P}_{sys}$ defines the LLM's role as an adaptive objective designer, aligning outputs with solver-compatible templates; (2) $\mathcal{P}_{geo}$ encodes geospatial semantics through Manhattan / Chicago zone graphs and OD matrices; (3) $\mathcal{P}_{model}$ provides class blueprints to enable variable-aware objective formulation. In contrast to $\mathcal{P}_{data}$, which merely specifies input formats or argument types, $\mathcal{P}_{model}$ ensures LLM-generated objectives respect both class structure and solver constraints. Further details are provided in Appendix A.9.1.

### 3.2.2 Dynamic environment feedback

Since each simulation run involves multiple decision-making steps, there are two approaches to enabling inference with an LLM, as captured in Figure 2: (1) a one-time query at the beginning of the test, where the generated objective remains fixed for all subsequent steps, or (2) queries at each step of the test. These correspond to two inference strategies. From a control perspective,

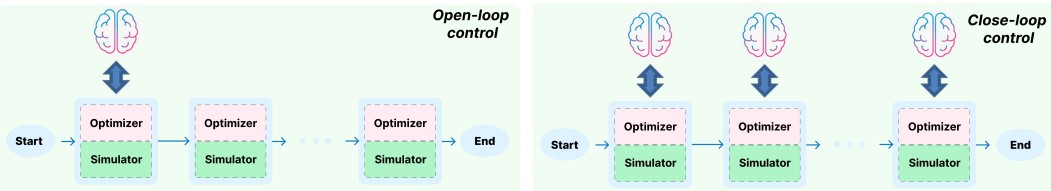

Figure 2: Inference strategies in a single simulation run: (1) Open-loop control, where a one-time query occurs at the beginning of the test. (2) Closed-loop control, where queries occur at each step of the test.

the second approach, where multiple queries occur, incorporates real-time environment feedback from the simulator, allowing the LLM to generate updated objectives dynamically. This forms a closed-loop control system, which is generally more effective for adaptive decision making. Thus, we adopt this approach in our inference strategy. To implement the closed-loop approach, the latest environment state must be incorporated into the prompt, ensuring the LLM remains context-aware. Specifically, we define the prompt text as follows.

*Dynamic states streaming* $\mathcal{P}_{dyn}$: At time step $t$, the prompt context $\mathcal{P}_{dyn}^t = \{(\mathbf{veh}_t, \mathbf{pass}_t)\}$, where $\mathbf{veh}_t \in \mathbb{R}^{|veh| \times 2}$ denotes vehicle state vector, such as positions and arrival time. $\mathbf{pass}_t \in \mathbb{R}^{|pass| \times 3}$ denotes demand tensor, including origin-destination pairs and request time.

### 3.2.3 Evolutionary prompt optimization

To facilitate reasoning in the LLM through experiential learning, we iteratively optimize a population of prompts using an evolutionary algorithm (EA). Departing from traditional EAs where individuals are parameter vectors, each prompt here is treated as an evolving entity, with fitness evaluated by simulation-based inference scores. Prompt evolution is guided by heuristics generated via the harmony search (HS) algorithm, a method well-established in classical optimization.

**Harmony search algorithm** HS emulates the process of musical performance, aiming to achieve an optimal state of harmony. As depicted in Algorithm 1, its evolution is governed by two key parameters: harmony memory considering rate (HMCR) and pitch adjustment rate (PAR). HMCR determines whether the next iteration involves an exploration branch (introducing new solutions) or an exploitation branch (refining existing solutions). If exploitation is selected, PAR facilitates local adjustments with a certain probability, allowing fine-tuned modifications. The prompt evolution process is formalized through a population matrix $\mathcal{P} \in \mathbb{R}^{I \times N \times T}$ where $I, N, T$ denote iteration times, population size and simulation steps.

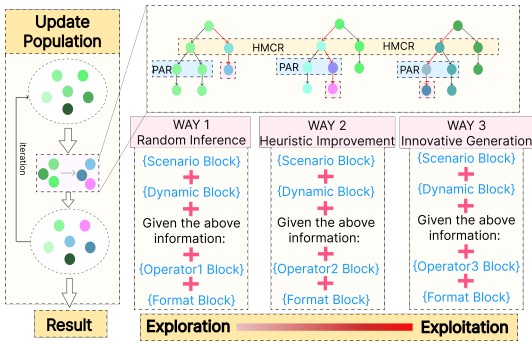

Figure 3: Evolutionary mechanism of harmony search algorithm

$\mathcal{P}_i = [Y_1^{(i)}, ..., Y_N^{(i)}]$, $Y_k^{(i)} = [X_k^{(i)}, f_k^{(i)}]$, here $X_k^{(i)}$ denotes the k-th individual at iteration $i$, $f_k^{(i)}$ is its fitness value, also, $X_k^{(i)} = [x_k^{(i)}(1), ..., x_k^{(i)}(T)]$. The fitness landscape is defined as: $f(X) = \mathbb{E}_{\tau \sim \mathcal{D}}[\mathcal{R}(\mathcal{E}(X, \tau))]$, where $\mathcal{E}$ represents the simulation environment and $\mathcal{R}$ the fitness cost over trajectory $\tau$. We adapt the HS algorithm through a probabilistic transition matrix:

$$\Pi = \begin{cases} \text{Exploration} & \text{if } \alpha > \text{HMCR} \\ \text{Exploitation} & \text{if } \alpha \leq \text{HMCR} \end{cases} \tag{2}$$

where $\alpha \sim U(0, 1)$. The exploitation phase incorporates dual refinement strategies: $W_{\text{exploit}} = \lambda W_{\text{heuristic}} + (1 - \lambda) W_{\text{innovative}}$ with $\lambda$ controlled by PAR.

As illustrated in Figure 3, varying the value of $\alpha$ and $\lambda$ results in the selection of one of three distinct prompt compositions. Each composition consists of four components: a scenario block $\mathcal{P}_{sce}$, a dynamic constraint block $\mathcal{P}_{dyn}$, an operator block defined in Table 1, and a format restriction block $\mathcal{P}_{restriction}$, with further implementation details provided in Appendix A.9.3.

Table 1: Three operators for LLM-driven objective generation. Equations and example prompts illustrate the mechanisms of each operator.

| *Random Inference* ($W_1$) | *Heuristic Improvement* ($W_2$) | *Innovative Generation* ($W_3$) |
| --- | --- | --- |
| If exploration is chosen, the LLM generates a new objective without historical knowledge. $X_{\text{new}} = \text{LLM}(\emptyset, \mathcal{P}_{\text{sce}})$ | The LLM generates an enhanced prompt by refining heuristics derived from the parent prompt. $X_{\text{new}} = \text{LLM}(X_{\text{parent}}, \mathcal{P}_{\text{sce}}, \mathcal{I}_1)$ | The LLM formulates a novel prompt, drawing inspiration from the parent prompt. $X_{\text{new}} = \text{LLM}(X_{\text{parent}}, \mathcal{P}_{\text{sce}}, \mathcal{I}_2)$ |
| Example prompt: `"Please generate a new objective for first-level assignment model."` | Example prompt: `"Develop an improved objective function by [instruction block1]."` | Example prompt: `"Reinvent the objective function from the previous run by [instruction block2]."` |

Our three evolutionary operators are listed in Table 1. The instruction blocks "[instruction block1]" and "[instruction block2]" contain specific directives assigned to their respective operators, generated by DeepSeek-R1 [37][38]. A detailed explanation is provided in Appendix A.9.2.

---

**Algorithm 1** Generate New Individual

---

**Require:** HMCR, PAR, Simulation steps $t \in T$, Population size $N$, Population $\mathcal{P}_0 = \{(X_1, fitness_1), ..., (X_N, fitness_N)\}$

**Ensure:** $NewInd_t$ for $t \in T$

1: **for** step $t = 1$ **to** $T$ **do**
2:     **if** rand1() > HMCR **then**
3:         $NewInd_t \leftarrow \{W_1\}$
4:     **else**
5:         $(X_i, fitness_i) \leftarrow \text{INDIVIDUALSELECT}(\mathcal{P})$
6:         $\tilde{X}_i \leftarrow \text{TOKENSELECT}(X_i)$
7:         $\lambda \sim \text{Bernoulli}(PAR)$
8:         $NewInd_t \leftarrow \lambda W_2(\tilde{X}_i, f_i) + (1 - \lambda)W_3(\tilde{X}_i, f_i)$
9:     **end if**
10: **end for**

---

Multiple queries are issued at different time steps throughout a complete simulation run, thus, the operator categories at different time steps are selected collectively in Algorithm 1. At this stage, only the parent prompt $(X_i, fitness_i)$ and operator category are chosen, while additional components, such as the scenario block, dynamic block, and format block, are incorporated to construct a complete prompt when the simulation step is executed. A detailed full loop algorithm of this process is provided in Appendix A.4. In Algorithm 1, function INDIVIDUALSELECT is trying to select an elite parent (individual with good fitness) from the old population. Each individual consists of three components: dynamic environment inputs spanning $T$ time steps; corresponding LLM response objectives, which also span $T$ time steps; and the final fitness value evaluated over a horizon of $T$. As the horizon $T$ increases, the context length for each parent becomes substantially larger. To mitigate the growing token size, the function TOKENSELECT is employed to retain only the most critical components, specifically the objective part, which is then combined with the fitness value to construct the final parent prompt.

## 4 Experiments

### 4.1 Experiment Settings

**Dataset and baselines** The 9 testing scenarios in Table 2 are constructed using the New York taxi dataset [39]. We also test on Chicago taxi dataset [40], experiments and details are provided in Table 3 at Appendix 4.2. A comprehensive description of the scenario generation process is presented in Appendix A.3. The details of all baseline methods are provided in Appendix A.2.

**Evaluation metrics** In the simulator, the waiting time for each passenger is computed, and the mean waiting time is used as the evaluation metric, as presented in Tables 2 and 3. Accordingly, methods employing two-level optimization frameworks should consider passenger waiting time as an objective in the second-level sequencing problem. Similarly, in the RL approach, minimizing the total time serves as the reward signal for model training. For LLM-based methods, the prompt is provided to instruct the LLM to minimize passenger waiting time.

**Implementation details** In our experimental setup, we utilize the DeepSeek-R1-Distill-Qwen-32B [41] model through the Hugging Face platform API as the default large language model for all LLM-based methods, which allow us to evaluate the adaptability of our method on smaller LLMs, thereby highlighting its potential applications. The temperature parameter is configured to 0.9. LLM-based methods all executed 3 times for each scenario, and the mean value of these three runs is reported in Tables 2 and 3. FunSearch is performed under 20 iterations. EOH and our method all employ 10 iterations with a population size of 5. All optimizer-based methods, either manual objectives or our adaptive-objective method, optimization solver Gurobi [42] is adopted to solve the problem running on a PC with 13th Gen Intel Core i9-13900KF × 32 CPU up to 5.80 GHz and RAM 32GB.

### 4.2 Experiment Comparison

**Baseline comparisons on Manhattan taxi dataset** Table 2 presents the experimental results on Downtown Manhattan. Our method outperforms the best baseline (FunSearch) in 7 out of 9 scenarios, excluding small-scale cases 1 and 3, and achieves an average improvement exceeding 14% for all

scenarios. In large-scale scenarios (8 and 9), it further reduces mean passenger delay by over 40% compared to the strongest baseline (Full RL). Three key findings can be found:

- Manual objective limitations: While composite objective functions that integrate distance, temporal, and utilization signals (Distance $\times$ Temporal $\times$ Utilization) outperform single-objective variants (e.g., 5.77 min vs. 14.88 min in Scenario 3), their effectiveness deteriorates under increased problem scale: Waiting times increase to 9.27 min in Scenario 9 (P200_-C100_T1200), indicating limited scalability.

- RL-based and LLM-only methods: RL performs well in low-complexity settings (1.27 min and 3.10 min in Scenario 1) but struggles with reward sparsity and exploration inefficiencies in larger scenarios, especially the first RL method (10.35 min in Scenario 9). While LLM-based EoH struggles with dynamic adaptation (10.27 min in Scenario 9), Funsearch's heuristics show better generalization (8.98 min). However, both approaches lack dynamic, per-time-interval feedback and a low-level optimizer to enforce solution quality, resulting in suboptimal performance compared to our hybrid method in large cases.

- Hybrid LLM+optimizer approach: By coupling LLM adaptability with optimizer precision, our hybrid framework delivers consistent performance gains, particularly under high-demand, long-horizon conditions (4.01 min in Scenario 9), achieving approximately 40% improvement over Full RL (6.82 min). The performance gap widens with scale, highlighting the importance of dynamic objective formulation and closed-loop LLM-optimizer interaction.

Our approach succeeds by combining LLM-driven semantic reasoning with solver-enforced mathematical rigor. LLM iteratively refines high-level objectives through prompt-based updates, overcoming the rigidity of static objectives in traditional systems that lack low-level constraint awareness. Meanwhile, solvers ensure spatiotemporal feasibility and numerical precision at scale without relying on large training datasets, as required in RL. This closed-loop mechanism, adapting objectives via the LLM and enforcing feasibility via the solver, effectively balances exploration and tractability, achieving state-of-the-art performance on mobility-on-demand benchmarks. Further analysis and experiments on Chicago dataset are provided in Appendix A.1.1.

Table 2: Average passenger waiting time (minutes) across optimization methods and scenarios on New York taxi dataset

| Category | Method | Scenario | | | | | | | | |
|---|---|---|---|---|---|---|---|---|---|---|
| | | 1 | 2 | 3 | 4 | 5 | 6 | 7 | 8 | 9 |
| Manual Objectives | Distance[†] [2] | 6.51 | 14.78 | 14.88 | 11.42 | 24.67 | 22.32 | 18.09 | 34.98 | 28.04 |
| | Distance $\times$ Utilization* | 5.02 | 7.38 | 7.94 | 6.96 | 9.07 | 9.89 | 7.06 | 9.24 | 11.59 |
| | Temporal[‡] $\times$ Utilization* [24] | 10.09 | 8.81 | 9.93 | 9.74 | 11.48 | 15.23 | 10.97 | 13.26 | 18.99 |
| | Distance $\times$ Temporal $\times$ Utilization | 3.32 | 4.99 | 5.77 | 4.37 | 6.06 | 6.72 | 5.19 | 7.07 | 9.27 |
| RL Methods | Default* + RL-Seq[◇] [23] | 3.10 | 4.37 | 4.90 | 4.92 | 6.63 | 6.59 | 7.48 | 7.51 | 10.35 |
| | Full RL [6] | 1.27 | 2.35 | 3.98 | 2.42 | 3.80 | 4.83 | 3.20 | 4.78 | 6.82 |
| LLM Methods | FunSearch [15] | **0.77** | 1.74 | **1.55** | 5.29 | 2.95 | 5.43 | 5.78 | 5.79 | 10.27 |
| | EoH [16] | 1.64 | 1.58 | 2.79 | 3.04 | 4.68 | 5.71 | 3.87 | 5.55 | 8.98 |
| **Hybrid (LLM+Optimizer)** | **Ours** | 1.55 | **1.37** | 2.50 | **1.89** | **2.59** | **4.14** | **3.10** | **2.25** | **4.01** |

Distance: Travel time between vehicle start position and passenger pickup&dropoff location.
Temporal: Gap between the vehicle non-idle time and passenger request time.
Utilization: Taxi service efficiency (vehicles/request).
Default: Use default objective (Distance $\times$ Temporal $\times$ Utilization) in first-level assignment optimization.
RL-Seq: Reinforcement learning method is adopted to solve second-level sequencing problem.

**Baseline comparisons on Chicago taxi dataset**   Due to the relatively larger spatial extent of Chicago's zones compared to Manhattan, combined with the highly imbalanced distribution of ride requests (Appendix A.3 for details), the average origin-destination (OD) travel times in Chicago are significantly longer. This increased travel distance contributes to overall higher passenger waiting times across the scenarios in Table 3. Table 3 demonstrates our framework's consistent superiority on Chicago distribution conditions, reducing mean passenger waiting times by an average of 18% in large-scale scenarios (e.g., 10.79 min vs. 14.35 min in Scenario 7, 14.51 min vs. 15.93 min in Scenario 8, 15.37 min vs. 19.79 min in Scenario 9). Compared to the best baseline (FunSearch), our method outperforms in 8 of 9 scenarios, with a marginal exception in small-scale case 3, and delivers an average improvement exceeding 18% across all cases. Three critical insights can be found:

- Manual objective limitations: Composite objectives (Distance × Temporal × Utilization) degrade severely under scale, with delays escalating to 25.01 min in Scenario 9 (vs. 15.37 min for our proposed approach). When the trip distribution is significantly imbalanced, as shown in Appendix A.3, single-objective variants that optimize solely for travel distance perform poorly under high-demand conditions, resulting in extreme delays (145.14 min in Scenario 8).

- RL-based and LLM-only methods: Full RL outperforms Default+RL-Seq (20.03 min vs. 24.63 min) but remains inferior to our proposed methods, with a gap of 4.66 min compared to the best-performing approach (15.37 min). This underscores RL's sensitivity to reward design and training data coverage. LLM-only methods exhibit inconsistent adaptation. Specifically, FunSearch outperforms EoH in smaller-scale scenarios (Scenarios 1–3), whereas EoH achieves better results in larger-scale settings (Scenarios 7–9). However, both methods lack dynamic feedback mechanisms and a fine-grained optimization layer to ensure solution quality, leading to inferior performance relative to our proposed hybrid approach.

- Hybrid LLM+optimizer approach: Our framework delivers robust performance across all scenarios, with several cases exhibiting substantial gains. For instance, in Scenario 4, our method achieves an average delay of 8.40 min compared to 12.05 min with the Full RL; in Scenario 7, 10.79 min vs. 14.35 min with the Full RL; and in Scenario 9, 15.37 min compared to 19.79 min with EoH. These improvements, all exceeding 20%, underscore the importance of closed-loop adaptation in dynamic and high-demand environments.

These results generalize the findings from the Manhattan dataset (Table 2), proving our method's adaptability to varying urban layouts and demand distributions. The advantage in large-scale scenarios underscores the necessity of iterative LLM-optimizer interaction for real-world ride-hailing systems.

Table 3: Average passenger waiting time (minutes) across optimization methods and scenarios on Chicago taxi dataset

| Category | Method | Scenario | | | | | | | | |
|---|---|---|---|---|---|---|---|---|---|---|
| | | 1 | 2 | 3 | 4 | 5 | 6 | 7 | 8 | 9 |
| Manual Objectives | Distance[†] [2] | 17.52 | 35.87 | 47.85 | 41.08 | 80.96 | 121.41 | 49.99 | 145.14 | 102.44 |
| | Distance × Utilization[*] | 11.60 | 15.34 | 18.35 | 15.70 | 18.50 | 22.77 | 19.07 | 22.74 | 30.38 |
| | Temporal[‡] × Utilization[*] [24] | 17.07 | 18.15 | 16.90 | 17.33 | 19.74 | 23.96 | 18.51 | 22.09 | 26.96 |
| | Distance × Temporal × Utilization | 9.54 | 13.79 | 18.19 | 11.81 | 16.71 | 20.74 | 17.21 | 20.68 | 25.01 |
| RL Methods | Default[*] + RL-Seq[◇] [23] | 10.94 | 10.87 | 15.37 | 19.27 | 18.68 | 21.57 | 24.02 | 24.86 | 24.63 |
| | Full RL [6] | 10.83 | 13.43 | 15.40 | 12.05 | 14.43 | 16.50 | 14.35 | 15.93 | 20.03 |
| LLM Methods | FunSearch [15] | 9.03 | 10.70 | **10.87** | 21.70 | 12.82 | 15.35 | 18.29 | 17.42 | 21.74 |
| | EoH [16] | 10.43 | 10.83 | 13.05 | 13.53 | 14.45 | 17.16 | 15.01 | 16.93 | 19.79 |
| **Hybrid (LLM+Optimizer)** | **Ours** | **8.65** | **9.58** | 11.30 | **8.40** | **12.32** | **14.96** | **10.79** | **14.51** | **15.37** |

Distance: Travel time between vehicle start position and passenger pickup&dropoff location.
Temporal: Gap between the vehicle non-idle time and passenger request time.
Utilization: Taxi service efficiency (vehicles/request).
Default: use default objective (Distance × Temporal × Utilization) in first-level assignment optimization.
RL-Seq: reinforcement learning method is adopted to solve second-level sequencing problem.

### 4.3 Ablation Study

**Scenario prompt composition** Incorporating $\mathcal{P}_{\text{model}}$, which encodes a structural blueprint of the optimization model, significantly reduces waiting times (e.g., 2.86±0.29 min for P70_C60_T600), though variability increases due to

Table 4: Average waiting time (minutes) by prompt composition

| Prompt Composition | P50_C30_T300 | P70_C60_T600 | P100_C80_T900 | P130_C80_T1200 |
|---|---|---|---|---|
| $\mathcal{P}_{\text{sys}} \cup \mathcal{P}_{\text{geo}} \cup \mathcal{P}_{\text{data}}$ | $9.93 \pm 0.00$ | $5.86 \pm 0.00$ | $5.29 \pm 0.00$ | $6.46 \pm 0.00$ |
| $\mathcal{P}_{\text{sys}} \cup \mathcal{P}_{\text{geo}} \cup \mathcal{P}_{\text{model}}$ | $8.72 \pm 0.91$ | $\mathbf{2.86 \pm 0.29}$ | $3.07 \pm 0.89$ | $4.54 \pm 0.36$ |
| $+\mathcal{P}_{\text{restriction}}$ | $\mathbf{8.16 \pm 1.48}$ | $4.12 \pm 0.83$ | $\mathbf{2.59 \pm 0.52}$ | $\mathbf{2.25 \pm 0.05}$ |

*Note*: Mean ± standard deviation across runs. Bold: best per scenario (P=Passengers, C=Taxis, T=Time(s)). Prompt variants: - $\mathcal{P}_{\text{sys}} \cup \mathcal{P}_{\text{geo}} \cup \mathcal{P}_{\text{data}}$: System+Geo+Data structure (Sec. 3.2.1). - $\mathcal{P}_{\text{sys}} \cup \mathcal{P}_{\text{geo}} \cup \mathcal{P}_{\text{model}}$: System+Geo+Model structure (Sec. 3.2.1). - $+\mathcal{P}_{\text{restriction}}$ denotes $\mathcal{P}_{\text{sys}} \cup \mathcal{P}_{\text{geo}} \cup \mathcal{P}_{\text{model}} \cup \mathcal{P}_{\text{restriction}}$: System+Geo+Model+Gurobi-compatible constraints (App. A.9.3). $\mathcal{P}_{\text{data}}$ defines variable formats, $\mathcal{P}_{\text{model}}$ specifies full MILP structure.

LLM-generated function diversity. Further integrating $\mathcal{P}_{\text{restriction}}$ (Gurobi-compatible constraints) achieves the lowest costs in three compositions (e.g., 2.25±0.05 min for P130_C80_T1200).

Table 4 compares the performance of three prompt compositions involving scenario block and format block, with means and standard deviations computed over three runs (visualized in Figure 4). When only basic system parameters and data specifications are provided, the model produces high passenger waiting times (e.g., 9.93±0.00 min for P50_C30_T300) with zero standard deviation across runs. This consistency is attributed to the generation of invalid objective functions, which are rejected during Gurobi's model verification; as a fallback, a static default objective is applied (see Table 6 in the Appendix).

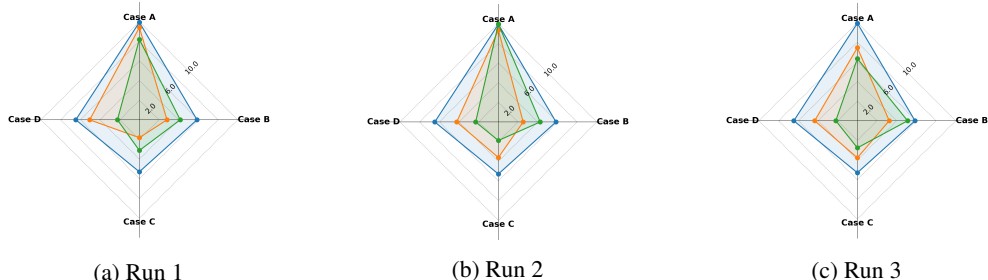

| (a) Run 1 | (b) Run 2 | (c) Run 3 |

Figure 4: Average passenger waiting time (minutes) under different compositions and runs. Cases: A (P50_C30_T300), B (P70_C60_T600), C (P100_C80_T900), and D (P130_C80_T1200). Blue line ($\mathcal{P}_{\text{sys}} \cup \mathcal{P}_{\text{geo}} \cup \mathcal{P}_{\text{data}}$), orange line ($\mathcal{P}_{\text{sys}} \cup \mathcal{P}_{\text{geo}} \cup \mathcal{P}_{\text{model}}$), green line ($\mathcal{P}_{\text{sys}} \cup \mathcal{P}_{\text{geo}} \cup \mathcal{P}_{\text{data}} \cup \mathcal{P}_{\text{restriction}}$).

Figure 4 illustrates the average passenger waiting time corresponding to Table 4. When problem scale is small (Case A and Case B), $\mathcal{P}_{\text{model}}$ is enough, but as complexity increases, $\mathcal{P}_{\text{restriction}}$ becomes essential. This aligns with Table 6, where larger scenarios exhibit higher error rates for compositions lacking constraints (e.g., 36.7% errors in P130_C80_T1200 without $\mathcal{P}_{\text{restriction}}$), necessitating explicit formalization to stabilize LLM outputs. This underscores the necessity of structured task grounding (model definitions) and constraint formalization for reliable LLM-driven optimization.

**Dynamic environment feedback impacts** As shown in Table 5, we perform experiments on two query mechanisms - single time on first time step and multiple times on each time step - at scenario P130_C80_T1200, the final result of each run at last iteration is provided. Clearly, the dynamic multi-time query demonstrates superior performance compared to the single-time query. The dynamic approach benefits from frequent interactions with the LLM, allowing it to acquire rich semantic information at each step. In contrast, the single query only provides the LLM with the initial environment setup, limiting its ability to propose an objective based on evolving information. This results in the generation of identical final outcomes, highlighting the limited flexibility of the single-query approach. Further discussion of cost reduction is provided in Figure 9 in Appendix A.1.1.

Table 5: Average passenger waiting time comparison: open-loop vs. dynamic closed-loop query mechanisms

| Query Mechanism | Run1 | Run2 | Run3 | Avg. |
|---|---|---|---|---|
| Single open-loop | 4.62 | 4.62 | 4.62 | 4.62 |
| Dynamic closed-loop | **2.23** | **2.31** | **2.19** | **2.24** |

*Note*: Values in minutes. Bold entries highlight the superior performance of the dynamic closed-loop mechanism.

## 5 Conclusion

In this paper, to solve the dynamic dispatching problem in a ride-hailing system, we propose a hybrid approach that combines LLM and optimizer in a dynamic hierarchical system, where LLM-generated objective in the high-level model is served as guiding heuristics for the low-level routing optimizer. High-level objectives are iteratively refined in a closed-loop evolutionary process, with performance evaluated across simulation epochs. A harmony search algorithm adaptively explores the LLM prompt space to improve objective quality over time. Experiments on Manhattan downtown and Central Chicago taxi datasets demonstrate the effectiveness of our approach, achieving an average 16% improvement over state-of-the-art baselines. Additional results, future work, and supporting materials are provided in Appendix A. The source code can be found in: `https://github.com/yizhangele/llm-guided-mod-optimization`.

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

# NeurIPS Paper Checklist

1. **Claims**

   Question: Do the main claims made in the abstract and introduction accurately reflect the paper's contributions and scope?

   Answer: [Yes]

   Justification: The abstract and introduction accurately reflect the paper's core contributions and scope. They clearly state the main methodological innovation: a hybrid framework that integrates LLM with mathematical optimization for dynamic decision-making in ride-hailing platforms. The abstract and introduction appropriately highlights the limitations of existing RL and decomposed optimization methods. They also outlines key components such as the training-free nature of the method, prompt-based high-level objective generation, and the closed-loop refinement mechanism via harmony search. These contributions are thoroughly developed in the main paper (Section 3.1 and Section 3.2 and Appendix Algorithm A.4) and evaluated through experiments (Section 4, Appendix A.1.1) on real-world datasets (New York and Chicago taxi data). The claims of performance improvements and semantic-guided optimization are substantiated by the results, making the abstract and introduction a faithful summary of the paper's contributions.

   Guidelines:

   - The answer NA means that the abstract and introduction do not include the claims made in the paper.
   - The abstract and/or introduction should clearly state the claims made, including the contributions made in the paper and important assumptions and limitations. A No or NA answer to this question will not be perceived well by the reviewers.
   - The claims made should match theoretical and experimental results, and reflect how much the results can be expected to generalize to other settings.
   - It is fine to include aspirational goals as motivation as long as it is clear that these goals are not attained by the paper.

2. **Limitations**

   Question: Does the paper discuss the limitations of the work performed by the authors?

   Answer: [Yes]

   Justification: Appendix A.1.2 discusses several limitations and outlines directions for future work, reflecting a clear understanding of the boundaries of the current study. Specifically, Appendix A.1.2 acknowledge the limitation of simulation fidelity, noting that the current experiments do not incorporate fine-grained, real-world traffic dynamics such as congestion patterns, signal timing, or road disruptions. To address this, we propose integrating microscopic traffic simulators like SUMO to enable higher-fidelity evaluations under realistic operational conditions.

   Moreover, we recognize the static role of the LLM in the current framework: it is not fine-tuned and adapts only through prompt-level feedback. Thus, we propose future enhancements involving reinforcement learning techniques to enable co-adaptation between the LLM and the optimizer, treating the optimizer as a critic to fine-tune the LLM more efficiently.

   Finally, we identify the potential for quantum computing to overcome current limitations in scalability and solution efficiency. We envision integrating quantum-enhanced solvers into the framework to accelerate both data-driven reasoning (via LLM) and combinatorial optimization, allowing tighter interaction and real-time deployment.

   Guidelines:

   - The answer NA means that the paper has no limitation while the answer No means that the paper has limitations, but those are not discussed in the paper.
   - The authors are encouraged to create a separate "Limitations" section in their paper.
   - The paper should point out any strong assumptions and how robust the results are to violations of these assumptions (e.g., independence assumptions, noiseless settings, model well-specification, asymptotic approximations only holding locally). The authors

should reflect on how these assumptions might be violated in practice and what the implications would be.

- The authors should reflect on the scope of the claims made, e.g., if the approach was only tested on a few datasets or with a few runs. In general, empirical results often depend on implicit assumptions, which should be articulated.

- The authors should reflect on the factors that influence the performance of the approach. For example, a facial recognition algorithm may perform poorly when image resolution is low or images are taken in low lighting. Or a speech-to-text system might not be used reliably to provide closed captions for online lectures because it fails to handle technical jargon.

- The authors should discuss the computational efficiency of the proposed algorithms and how they scale with dataset size.

- If applicable, the authors should discuss possible limitations of their approach to address problems of privacy and fairness.

- While the authors might fear that complete honesty about limitations might be used by reviewers as grounds for rejection, a worse outcome might be that reviewers discover limitations that aren't acknowledged in the paper. The authors should use their best judgment and recognize that individual actions in favor of transparency play an important role in developing norms that preserve the integrity of the community. Reviewers will be specifically instructed to not penalize honesty concerning limitations.

3. **Theory assumptions and proofs**

   Question: For each theoretical result, does the paper provide the full set of assumptions and a complete (and correct) proof?

   Answer: [Yes]

   Justification: The paper includes a formal mixed-integer logic model formulation of the ride-hailing problem, which is introduced and discussed in the Appendix A.6. The conversion of logical constraints into equivalent mixed-integer linear constraints is also detailed, with accompanying proofs provided in Appendix A.7 to ensure correctness. While the main paper focuses on the hierarchical decomposition into high-level assignment and low-level routing problems, as part of the LLM-guided optimization loop, it clearly states the modeling assumptions and system-level structure. Additionally, the theoretical foundations behind the modular decomposition and prompt-based refinement via harmony search are supported by formal modeling and complete derivations in the Appendix.

   Guidelines:

   - The answer NA means that the paper does not include theoretical results.

   - All the theorems, formulas, and proofs in the paper should be numbered and cross-referenced.

   - All assumptions should be clearly stated or referenced in the statement of any theorems.

   - The proofs can either appear in the main paper or the supplemental material, but if they appear in the supplemental material, the authors are encouraged to provide a short proof sketch to provide intuition.

   - Inversely, any informal proof provided in the core of the paper should be complemented by formal proofs provided in appendix or supplemental material.

   - Theorems and Lemmas that the proof relies upon should be properly referenced.

4. **Experimental result reproducibility**

   Question: Does the paper fully disclose all the information needed to reproduce the main experimental results of the paper to the extent that it affects the main claims and/or conclusions of the paper (regardless of whether the code and data are provided or not)?

   Answer: [Yes]

   Justification: Section 4.1 presents the experimental setup, evaluation metrics, and implementation details. The New York and Chicago taxi datasets used in the experiments are described in Appendix A.3. Benchmark comparisons with baseline methods are provided in Table 2 and Figure 5 for the New York dataset (main paper), and in Table 3 and Figure 6 for the Chicago dataset.

Ablation studies are conducted to evaluate several key aspects of the system. The impact of scenario prompt composition is analyzed in Table 4 and Figure 4. The effect of dynamic environment feedback mechanisms is detailed in Table 5. Additional analyses presented in the appendix include: cost variations over iterations for both multi-query and single-query settings (Figure 9); an evolutionary hyperparameter study (Figures 10 and 11); an investigation of LLM temperature sensitivity (Figure 12); and an analysis of LLM-induced error rates (Table 6).

To preserve contribution integrity and ensure proper attribution, code is provided in `https://github.com/yizhangele/llm-guided-mod-optimization` to support future research.

Guidelines:

- The answer NA means that the paper does not include experiments.
- If the paper includes experiments, a No answer to this question will not be perceived well by the reviewers: Making the paper reproducible is important, regardless of whether the code and data are provided or not.
- If the contribution is a dataset and/or model, the authors should describe the steps taken to make their results reproducible or verifiable.
- Depending on the contribution, reproducibility can be accomplished in various ways. For example, if the contribution is a novel architecture, describing the architecture fully might suffice, or if the contribution is a specific model and empirical evaluation, it may be necessary to either make it possible for others to replicate the model with the same dataset, or provide access to the model. In general. releasing code and data is often one good way to accomplish this, but reproducibility can also be provided via detailed instructions for how to replicate the results, access to a hosted model (e.g., in the case of a large language model), releasing of a model checkpoint, or other means that are appropriate to the research performed.
- While NeurIPS does not require releasing code, the conference does require all submissions to provide some reasonable avenue for reproducibility, which may depend on the nature of the contribution. For example
  (a) If the contribution is primarily a new algorithm, the paper should make it clear how to reproduce that algorithm.
  (b) If the contribution is primarily a new model architecture, the paper should describe the architecture clearly and fully.
  (c) If the contribution is a new model (e.g., a large language model), then there should either be a way to access this model for reproducing the results or a way to reproduce the model (e.g., with an open-source dataset or instructions for how to construct the dataset).
  (d) We recognize that reproducibility may be tricky in some cases, in which case authors are welcome to describe the particular way they provide for reproducibility. In the case of closed-source models, it may be that access to the model is limited in some way (e.g., to registered users), but it should be possible for other researchers to have some path to reproducing or verifying the results.

5. **Open access to data and code**

   Question: Does the paper provide open access to the data and code, with sufficient instructions to faithfully reproduce the main experimental results, as described in supplemental material?

   Answer: [Yes]

   Justification: Our codebase is now publicly available, can be found in `https://github.com/yizhangele/llm-guided-mod-optimization`.

   Guidelines:

   - The answer NA means that paper does not include experiments requiring code.
   - Please see the NeurIPS code and data submission guidelines (`https://nips.cc/public/guides/CodeSubmissionPolicy`) for more details.
   - While we encourage the release of code and data, we understand that this might not be possible, so "No" is an acceptable answer. Papers cannot be rejected simply for not

including code, unless this is central to the contribution (e.g., for a new open-source benchmark).

- The instructions should contain the exact command and environment needed to run to reproduce the results. See the NeurIPS code and data submission guidelines (`https://nips.cc/public/guides/CodeSubmissionPolicy`) for more details.
- The authors should provide instructions on data access and preparation, including how to access the raw data, preprocessed data, intermediate data, and generated data, etc.
- The authors should provide scripts to reproduce all experimental results for the new proposed method and baselines. If only a subset of experiments are reproducible, they should state which ones are omitted from the script and why.
- At submission time, to preserve anonymity, the authors should release anonymized versions (if applicable).
- Providing as much information as possible in supplemental material (appended to the paper) is recommended, but including URLs to data and code is permitted.

6. **Experimental setting/details**

Question: Does the paper specify all the training and test details (e.g., data splits, hyper-parameters, how they were chosen, type of optimizer, etc.) necessary to understand the results?

Answer: [Yes]

Justification: While our work does not involve training machine learning models in the traditional sense, it integrates a well-established pretrained LLM with a mathematical optimization framework. As such, there are no training/test data splits or model training procedures to report. However, to ensure clarity of results, we provide: (1) Full implementation details for all components involved in our approach in Section 4.1 **Implementation details**; (2) The New York and Chicago taxi datasets, along with the methodology for generating the nine testing scenarios, are described in Appendix A.3; (3) To evolve prompt heuristics for the LLM, we utilize the harmony search algorithm. Key hyperparameters of this algorithm (e.g., HMCR and PAR) are analyzed via a sensitivity study presented in the Figures 10 and 11 at Appendix A.1.1. These details collectively provide a complete picture of the experiment setup required to understand the results.

Guidelines:

- The answer NA means that the paper does not include experiments.
- The experimental setting should be presented in the core of the paper to a level of detail that is necessary to appreciate the results and make sense of them.
- The full details can be provided either with the code, in appendix, or as supplemental material.

7. **Experiment statistical significance**

Question: Does the paper report error bars suitably and correctly defined or other appropriate information about the statistical significance of the experiments?

Answer: [Yes]

Justification: Our study does not involve training a data-driven model but instead evaluates the performance and stability of a hybrid framework integrating LLM-based prompt generation with optimization solvers. To demonstrate statistical significance and variability:

(1) We benchmark our method on two distinct real-world taxi datasets (New York and Chicago), providing evidence of adaptability to divergent trip distributions.

(2) We evaluate the impact of prompt composition on LLM inference stability. Specifically, Table 4 reports average passenger waiting times across four scenarios and three prompt variants, showing mean and standard deviation over 3 independent runs per setting. These error bars reflect variation in LLM outputs due to prompt randomness and inherent nondeterminism.

(3) We study LLM output validity through an error rate analysis (Table 6 in Appendix A.1.1), showing acceptance rates of LLM-generated objectives by the Gurobi solver across prompt types and scenarios. Each configuration is evaluated over three independent runs, and average error rates are reported.

All reported statistics clarify the source of variability (LLM generation stochasticity) and the method used (empirical evaluation over multiple independent runs). Standard deviations are explicitly provided, and tables/figures are cross-referenced in the main text.

Guidelines:

- The answer NA means that the paper does not include experiments.
- The authors should answer "Yes" if the results are accompanied by error bars, confidence intervals, or statistical significance tests, at least for the experiments that support the main claims of the paper.
- The factors of variability that the error bars are capturing should be clearly stated (for example, train/test split, initialization, random drawing of some parameter, or overall run with given experimental conditions).
- The method for calculating the error bars should be explained (closed form formula, call to a library function, bootstrap, etc.)
- The assumptions made should be given (e.g., Normally distributed errors).
- It should be clear whether the error bar is the standard deviation or the standard error of the mean.
- It is OK to report 1-sigma error bars, but one should state it. The authors should preferably report a 2-sigma error bar than state that they have a 96% CI, if the hypothesis of Normality of errors is not verified.
- For asymmetric distributions, the authors should be careful not to show in tables or figures symmetric error bars that would yield results that are out of range (e.g. negative error rates).
- If error bars are reported in tables or plots, The authors should explain in the text how they were calculated and reference the corresponding figures or tables in the text.

8. **Experiments compute resources**

Question: For each experiment, does the paper provide sufficient information on the computer resources (type of compute workers, memory, time of execution) needed to reproduce the experiments?

Answer: [Yes]

Justification: The paper provides sufficient details regarding the computational resources used in Section 4.1. Specifically, all optimizer-based methods were executed on a local machine equipped with a 13th Gen Intel Core i9-13900KF CPU (32 cores, up to 5.80 GHz) and 32 GB RAM. For LLM-based methods, the DeepSeek-R1-Distill-Qwen-32B model was accessed via the Hugging Face platform API, and relevant configuration parameters (e.g., temperature = 0.9) were reported. Each LLM-based method was executed three times per scenario, and the average results are presented. However, since these methods rely on third-party API calls, the end-to-end runtime is influenced by network latency and server load on the Hugging Face platform, making it difficult to precisely report wall-clock execution times. Nonetheless, the number of iterations, population size, and evaluation protocol are clearly stated, ensuring reproducibility of the experimental procedures.

Guidelines:

- The answer NA means that the paper does not include experiments.
- The paper should indicate the type of compute workers CPU or GPU, internal cluster, or cloud provider, including relevant memory and storage.
- The paper should provide the amount of compute required for each of the individual experimental runs as well as estimate the total compute.
- The paper should disclose whether the full research project required more compute than the experiments reported in the paper (e.g., preliminary or failed experiments that didn't make it into the paper).

9. **Code of ethics**

Question: Does the research conducted in the paper conform, in every respect, with the NeurIPS Code of Ethics `https://neurips.cc/public/EthicsGuidelines`?

Answer: [Yes]

Justification: The research focuses on algorithmic development and evaluation using established optimization benchmarks. It does not involve human participants or engage with applications that raise evident ethical concerns. Based on this, we affirm that the work complies fully with the NeurIPS Code of Ethics.

Guidelines:

- The answer NA means that the authors have not reviewed the NeurIPS Code of Ethics.
- If the authors answer No, they should explain the special circumstances that require a deviation from the Code of Ethics.
- The authors should make sure to preserve anonymity (e.g., if there is a special consideration due to laws or regulations in their jurisdiction).

10. **Broader impacts**

Question: Does the paper discuss both potential positive societal impacts and negative societal impacts of the work performed?

Answer: [No]

Justification: The paper emphasizes technical contributions and does not explicitly address broader societal impacts. Although the application domains (such as transportation systems) suggest potential positive societal benefits, these are not specifically articulated. Likewise, possible negative societal consequences are not discussed in the manuscript.

Guidelines:

- The answer NA means that there is no societal impact of the work performed.
- If the authors answer NA or No, they should explain why their work has no societal impact or why the paper does not address societal impact.
- Examples of negative societal impacts include potential malicious or unintended uses (e.g., disinformation, generating fake profiles, surveillance), fairness considerations (e.g., deployment of technologies that could make decisions that unfairly impact specific groups), privacy considerations, and security considerations.
- The conference expects that many papers will be foundational research and not tied to particular applications, let alone deployments. However, if there is a direct path to any negative applications, the authors should point it out. For example, it is legitimate to point out that an improvement in the quality of generative models could be used to generate deepfakes for disinformation. On the other hand, it is not needed to point out that a generic algorithm for optimizing neural networks could enable people to train models that generate Deepfakes faster.
- The authors should consider possible harms that could arise when the technology is being used as intended and functioning correctly, harms that could arise when the technology is being used as intended but gives incorrect results, and harms following from (intentional or unintentional) misuse of the technology.
- If there are negative societal impacts, the authors could also discuss possible mitigation strategies (e.g., gated release of models, providing defenses in addition to attacks, mechanisms for monitoring misuse, mechanisms to monitor how a system learns from feedback over time, improving the efficiency and accessibility of ML).

11. **Safeguards**

Question: Does the paper describe safeguards that have been put in place for responsible release of data or models that have a high risk for misuse (e.g., pretrained language models, image generators, or scraped datasets)?

Answer: [NA]

Justification: The paper introduces a novel framework combining LLM with optimization techniques. The optimization model, its integration with LLM and the standard benchmark datasets employed do not present a high risk of misuse. As such, no additional release safeguards beyond standard open-source practices are deemed necessary.

Guidelines:

- The answer NA means that the paper poses no such risks.

- Released models that have a high risk for misuse or dual-use should be released with necessary safeguards to allow for controlled use of the model, for example by requiring that users adhere to usage guidelines or restrictions to access the model or implementing safety filters.
- Datasets that have been scraped from the Internet could pose safety risks. The authors should describe how they avoided releasing unsafe images.
- We recognize that providing effective safeguards is challenging, and many papers do not require this, but we encourage authors to take this into account and make a best faith effort.

12. **Licenses for existing assets**

Question: Are the creators or original owners of assets (e.g., code, data, models), used in the paper, properly credited and are the license and terms of use explicitly mentioned and properly respected?

Answer: [Yes]

Justification: The paper appropriately cites and credits all external assets, including baseline methods (e.g., Default + RL-Seq, FullRL, FunSearch, EoH) and datasets (New York taxi, Chicago taxi). Details are as follows:

- **Default + RL-Seq:** This method employs deep reinforcement learning for low-level sequencing. We have cited the relevant publication in our manuscript and utilized a publicly available implementation from GitHub. Although the repository does not explicitly specify a license, we have reached out to the repository's author to clarify the terms of use.
  `https://github.com/higgsfield/np-hard-deep-reinforcement-learning/tree/master`
- **FullRL:** This method applies reinforcement learning to solve the entire problem. The implementation is publicly accessible on GitHub. Although the repository does not specify a license, we have cited the associated publication in our manuscript and have contacted the author of Github repository to clarify usage permissions. We have received confirmation via email that our use of their code is permitted.
  `https://github.com/callmespring/MDPOD/tree/main.`
- **EoH:** The code is openly available on GitHub under the MIT license.
  `https://github.com/FeiLiu36/EoH/blob/main/LICENSE`
- **FunSearch:** We adopt the FunSearch implementation from the EoH baseline methods, which is publicly available and released under the Apache 2.0 license.
  `https://github.com/FeiLiu36/EoH/blob/main/baseline/funsearch/LICENSE`
- **New York Taxi Dataset:** This dataset is publicly available and maintained by the NYC Taxi and Limousine Commission. It is accessible via the NYC open data portal and is provided under the NYC Open Data Terms of Use, which generally allow use, modification, and redistribution for both commercial and non-commercial purposes, with attribution.
  `https://www.nyc.gov/site/tlc/about/tlc-trip-record-data.page`
  `https://opendata.cityofnewyork.us/overview/#termsofuse`
- **Chicago Taxi Dataset:** This dataset is publicly available via the city of Chicago data portal. While no specific license is listed, it is intended for public access and use. The city's data disclaimer outlines general terms that typically allow use, modification, and redistribution with appropriate attribution.
  `https://data.cityofchicago.org/Transportation/Taxi-Trips-2013-2023-/wrvz-psew/about_data`
  `https://www.chicago.gov/city/en/narr/foia/data_disclaimer.html`
- **DeepSeek-R1-Distill-Qwen-32B:** We utilize this publicly available large language model via the Hugging Face platform API for prompt optimization and LLM-based decision making. The model is released under the MIT License. DeepSeek-R1-Distill-Qwen-32B is derived from the Qwen-2.5 series (originally licensed under the Apache 2.0 License) and further fine-tuned on 800k curated samples by DeepSeek-R1. We cite the official model repository in our manuscript and adhere to all licensing terms.
  `https://huggingface.co/deepseek-ai/DeepSeek-R1-Distill-Qwen-32B`

Guidelines:

- The answer NA means that the paper does not use existing assets.
- The authors should cite the original paper that produced the code package or dataset.
- The authors should state which version of the asset is used and, if possible, include a URL.
- The name of the license (e.g., CC-BY 4.0) should be included for each asset.
- For scraped data from a particular source (e.g., website), the copyright and terms of service of that source should be provided.
- If assets are released, the license, copyright information, and terms of use in the package should be provided. For popular datasets, `paperswithcode.com/datasets` has curated licenses for some datasets. Their licensing guide can help determine the license of a dataset.
- For existing datasets that are re-packaged, both the original license and the license of the derived asset (if it has changed) should be provided.
- If this information is not available online, the authors are encouraged to reach out to the asset's creators.

13. **New assets**

    Question: Are new assets introduced in the paper well documented and is the documentation provided alongside the assets?

    Answer: [Yes]

    Justification: Our codebase accompanied with comprehensive documentation is now available in `https://github.com/yizhangele/llm-guided-mod-optimization`.

    Guidelines:

    - The answer NA means that the paper does not release new assets.
    - Researchers should communicate the details of the dataset/code/model as part of their submissions via structured templates. This includes details about training, license, limitations, etc.
    - The paper should discuss whether and how consent was obtained from people whose asset is used.
    - At submission time, remember to anonymize your assets (if applicable). You can either create an anonymized URL or include an anonymized zip file.

14. **Crowdsourcing and research with human subjects**

    Question: For crowdsourcing experiments and research with human subjects, does the paper include the full text of instructions given to participants and screenshots, if applicable, as well as details about compensation (if any)?

    Answer: [NA]

    Justification: The research does not involve crowdsourcing experiments or research with human subjects.

    Guidelines:

    - The answer NA means that the paper does not involve crowdsourcing nor research with human subjects.
    - Including this information in the supplemental material is fine, but if the main contribution of the paper involves human subjects, then as much detail as possible should be included in the main paper.
    - According to the NeurIPS Code of Ethics, workers involved in data collection, curation, or other labor should be paid at least the minimum wage in the country of the data collector.

15. **Institutional review board (IRB) approvals or equivalent for research with human subjects**

    Question: Does the paper describe potential risks incurred by study participants, whether such risks were disclosed to the subjects, and whether Institutional Review Board (IRB) approvals (or an equivalent approval/review based on the requirements of your country or institution) were obtained?

Answer: [NA]

Justification: The research does not involve human subjects, therefore IRB approval is not applicable.

Guidelines:

- The answer NA means that the paper does not involve crowdsourcing nor research with human subjects.
- Depending on the country in which research is conducted, IRB approval (or equivalent) may be required for any human subjects research. If you obtained IRB approval, you should clearly state this in the paper.
- We recognize that the procedures for this may vary significantly between institutions and locations, and we expect authors to adhere to the NeurIPS Code of Ethics and the guidelines for their institution.
- For initial submissions, do not include any information that would break anonymity (if applicable), such as the institution conducting the review.

16. **Declaration of LLM usage**

Question: Does the paper describe the usage of LLMs if it is an important, original, or non-standard component of the core methods in this research? Note that if the LLM is used only for writing, editing, or formatting purposes and does not impact the core methodology, scientific rigorousness, or originality of the research, declaration is not required.

Answer: [Yes]

Justification: The core contribution of our paper is the integration of LLM into an online optimization framework for ride-hailing platforms. LLMs are not just used for writing or formatting but play a central, original role in the methodology. Specifically, our proposed framework employs LLM as a meta-optimizer that generate high-level semantic heuristics to guide a lower-level mathematical optimizer responsible for real-time decision-making under operational constraints. This approach addresses key limitations in traditional reinforcement learning and decomposed optimization methods by removing the need for training data and enabling adaptive objective formulation.

The usage of the LLM is described in detail in Section 3.2, where we introduce the hybrid LLM+optimizer framework. The full closed-loop mechanism, including how prompts are refined using evolutionary feedback, is further detailed in Algorithm A.4. We also discuss our prompt engineering strategies in Appendix A.9. These components demonstrate a novel and non-standard use of LLMs that directly impacts the originality, methodological rigor, and performance of our system.

Guidelines:

- The answer NA means that the core method development in this research does not involve LLMs as any important, original, or non-standard components.
- Please refer to our LLM policy (`https://neurips.cc/Conferences/2025/LLM`) for what should or should not be described.

# A    Technical Appendices

## A.1    Discussion

### A.1.1    Further studies

**Spatiotemporal analysis**    To illustrate the spatial-temporal distribution of passenger waiting times across different methods, Figure 7 and Figure 8 present the results for Scenario 9 (as defined in Table 2 and Table 3) for Downtown Manhattan and Central Chicago, respectively, where 200 passenger requests are dispatched within a 1200-second time window and served by 100 taxis. To facilitate spatial interpretation of the study area, Figures 5 and 6 show the zone-based maps of Downtown Manhattan and extended Central Chicago,

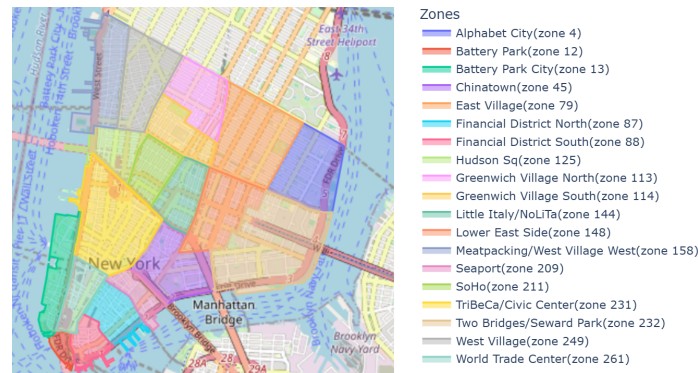

Figure 5: Zone-based Manhattan downtown map

respectively, with each study area partitioned into 19 predefined zones used as both origins and destinations for passenger trips.

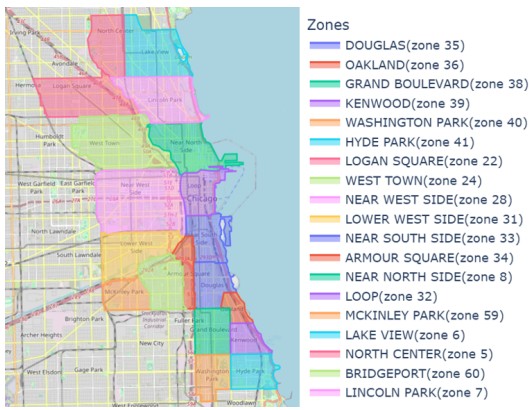

Figure 6: Zone-based Chicago extended central map

Compared to the zoning structure of Manhattan, Chicago's zones, also referred to community areas, are generally larger in spatial extent, especially outside the central business district. In particular, OD trip patterns reveal a notable contrast between the two cities: while Manhattan demonstrates a relatively balanced distribution of taxi trips across its zones, Chicago's taxi activity is highly concentrated within a few central areas, most prominently in zones 8, 32, and 28. Further details on the OD distribution are provided in Appendix A.3.

Figures 7 and 8 illustrate the spatiotemporal distribution of passenger waiting delays across zones in Downtown Manhattan and Central Chicago, respectively. Both figures correspond to the high-demand Scenario 9, as defined in Table 2 and Table 3. The passenger waiting delay is computed as $delay_p = max(t_p^{pick} - t_p^{req}, 0)$, where $t_p^{pick}$ denotes the vehicle pickup time and $t_p^{req}$ the passenger request time. Delays are aggregated into 600-second temporal bins (e.g., $t_p^{pick} \in [600s, 1200s]$ maps to Slot 1) and 800-second temporal bins (e.g., $t_p^{pick} \in [800s, 1600s]$ maps to Slot 1) for Figures 7 and 8, respectively. Based on the aggregated bins, delays are spatially averaged over a set of predefined geographic zones. This joint spatiotemporal aggregation captures localized service inefficiencies while preserving the dynamics of demand–supply imbalance over time. The heatmaps quantitatively validate our framework's advantage: under identical input conditions, our method completes all passenger pickups within 4 time slots (4×600 seconds for Figure 7, 4×800 seconds for Figure 8), whereas baseline methods require 5 to 7 time slots in the Manhattan setting and 5 to 9 time slots in the Chicago setting to fulfill all requests. As time progresses, the delays of unserved passengers accumulate, resulting in significantly higher average waiting times in the later time slots. This effect is reflected in the heatmap as progressively intensified red shading in the final temporal bins. As manual objective methods (Tables 2 and 3) often yield high passenger delays, only their best-performing approach (Distance × Temporal × Utilization) is included in Figures 7 and 8 for comparison.

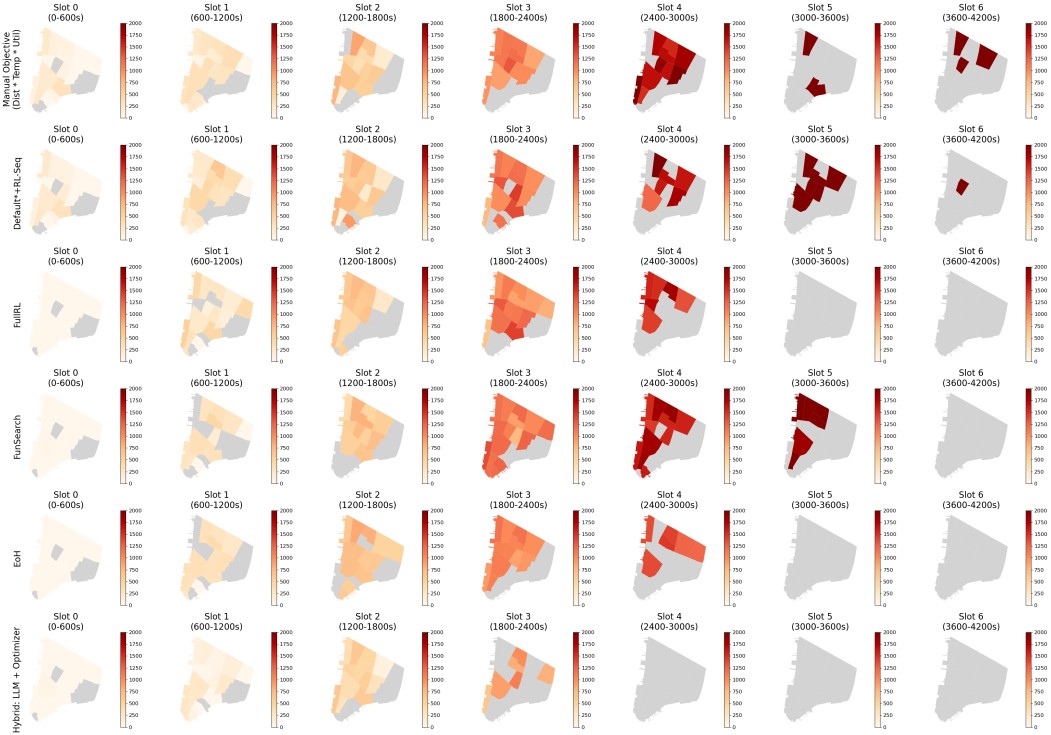

Figure 7: Spatial-temporal distribution of passenger waiting times across optimization methods in Scenario 9 (P200_C100_T1200) under New York taxi dataset. Each row represents a method: (1) Manual composite objective (Distance × Temporal × Utilization), (2) Default + RL-Seq, (3) Full RL, (4) FunSearch, (5) EoH, (6) Our LLM-Optimizer approach. Columns show 600-second time windows. Heatmap colors show average waiting times per zone.

In Figure 7, first approach (Row 1) exhibit persistent hotspots for Zones 144, 79 and 249, due to static designed objective. RL methods (row 2-3) show temporal degradation, especially row 2, where expanding high-delay regions in Slot 5 (3000–3600s) highlight RL's limitations in enforcing hard constraints (e.g., detour limits) and generalize to unseen demand patterns. LLM-based methods, notably EoH, perform moderately better, completing all requests within 5 time slots. Although FunSearch is the best overall baseline across scenarios, as it outperforms our method in Scenario 1 (0.77 min) and Scenario 3 (1.55 min) as shown in Table 2, its performance degrades significantly under high-demand conditions, particularly in Scenario 9, captured in Figure 7. In contrast, our proposed framework completes all requests within 4 time slots while maintaining spatially consistent low delays (light yellows) through closed-loop LLM-optimizer interaction. This analysis quantifies how our framework overcomes RL's data dependency, the rigidity of manually decomposed objectives, and the constraint unawareness of LLM-only methods. The heatmaps align with Scenario 9 results in Table 2, proving that semantic reasoning (LLMs) and symbolic grounding (solvers) effectively mitigate delay accumulation in large-scale urban mobility systems.

A similar pattern is observed in Figure 8. Unlike the results on the Manhattan dataset, the Chicago scenario requires more time to complete all requests, primarily due to a more imbalanced trip distribution and longer average travel times between origin-destination pairs. Specifically, the first (Row 1) and second (Row 2) baseline approaches require 9 and 8 time slots to serve all passengers, respectively. Full RL (Row 3) and EoH (Row 5) demonstrate improved performance, completing the task in 5 time slots, while FunSearch (Row 4) shows moderate degradation with 6 time slots. In contrast, the proposed method completes all requests within only 4 time slots, indicating enhanced adaptability to complex urban topologies and demand heterogeneity. This highlights the effectiveness of integrating LLM as a data-driven prior with mathematical optimization for robust and efficient decision-making.

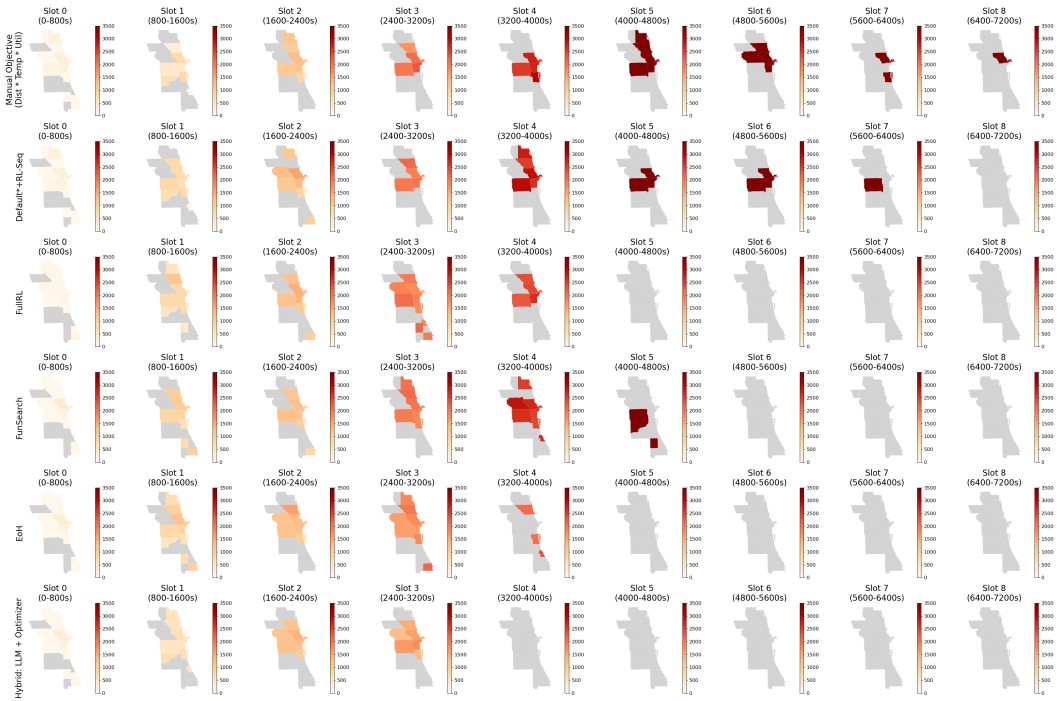

Figure 8: Spatial-temporal distribution of passenger waiting times across optimization methods in Scenario 9 (P200_C100_T1200) under Chicago taxi dataset. Each row represents a method: (1) Manual composite objective (Distance × Temporal × Utilization), (2) Default + RL-Seq, (3) Full RL, (4) FunSearch, (5) EoH, (6) Our LLM-Optimizer approach. Columns show 800-second time windows. Heatmap colors show average waiting times per zone.

**Query comparison** Figure 9 illustrates the cost changes under different iterations for multi-query and single-query, corresponding to the scenarios described in Table 5. Across all runs, multi-query achieves steeper convergence trajectories, reducing costs by an average of 50% (relative to single-query) within 10 iterations, which aligns with Table 5. This results from the closed-loop interaction between the LLM and the optimizer in the multi-query framework, in contrast to the open-loop configuration of the single-query approach.

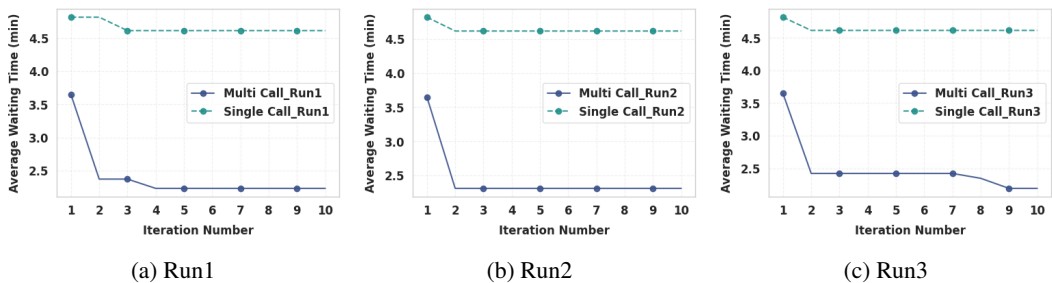

(a) Run1        (b) Run2        (c) Run3

Figure 9: Cost changes through iterations between multi-call and single-call approaches.

**Evolutionary hyperparameter study** As captured in Figure 10, a grid search is performed for the hyperparameters HMCR and PAR in the harmony search algorithm for P100_C80_T900. The HMCR values tested are 0.1, 0.5, and 0.9, while PAR values tested are 0.2, 0.5, and 0.8. Figure 10a and Figure 10b depict the best individual cost in last iteration and the mean cost of entire population in last iteration. The results demonstrate significant performance improvements under high HMCR (HMCR = 0.9), suggesting effective exploitation of high-quality parent solutions from memory. This

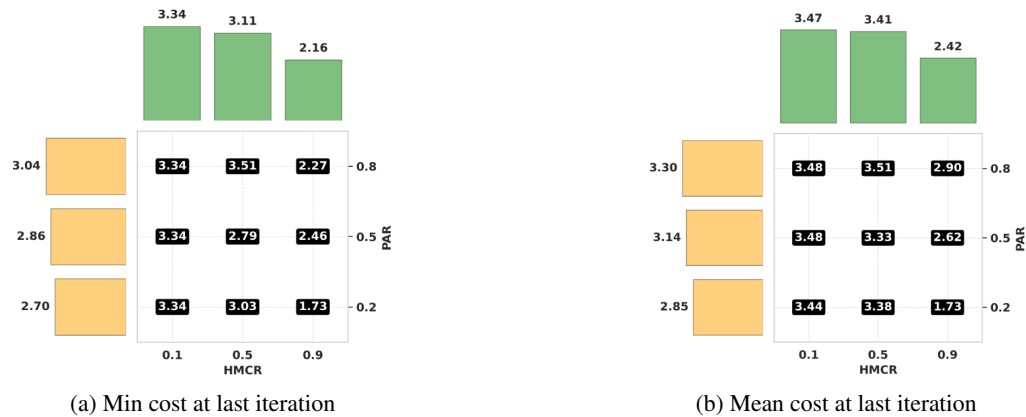

(a) Min cost at last iteration        (b) Mean cost at last iteration

Figure 10: Hyperparameter sensitivity analysis on HMCR and PAR - Cost at last iteration.

aligns with our hypothesis that preserving promising solution components through frequent memory recall (via operator $W_2$ and $W_3$) substantially enhances convergence properties. On the other hand, while lower PAR value (0.2) yield optimal results at HMCR = 0.9, this relationship no longer exists for HMCR $\leq 0.5$. Consequently, operator selection frequency between $W_2$ and $W_3$ appears governed more by solution quality feedback than fixed parameterization. A similar pattern in cost reduction is observed in Figure 11a, 11b and 11c, with a significant decrease in cost when HMCR is high.

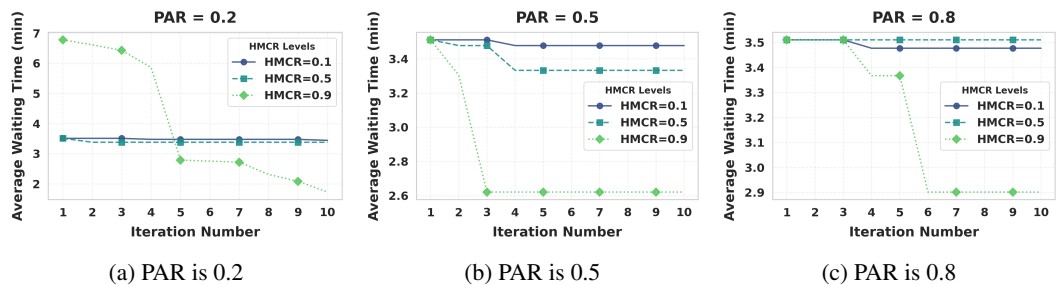

(a) PAR is 0.2        (b) PAR is 0.5        (c) PAR is 0.8

Figure 11: Hyperparameter sensitivity analysis on HMCR and PAR - Costs at various iterations.

**LLM temperature study** We evaluate the impact of LLM temperature ($\tau \in \{0.1, 0.5, 0.9, 1.3\}$) on optimization efficiency across three scenarios, as captured in Figure 12. The contour plots depict cost landscapes over 10 iterations and 4 temperatures, revealing that extreme temperatures ($\tau = 0.1$ and $\tau = 1.3$) consistently lead to higher overall costs, whereas $\tau = 0.9$ yields the lowest values, indicating more effective optimization. Across all settings, convergence occurs rapidly, with trajectories stabilizing within 2–3 iterations. When $\tau = 1.3$, high stochasticity disrupts objective function coherence, generating infeasible or unstable formulations. While $\tau = 0.1$ overly constrains diversity, causing premature convergence to suboptimal basins. The intermediate setting of $\tau = 0.9$ strikes an effective balance between exploitation (leveraging high-likelihood tokens for constraint adherence) and exploration (sampling novel objective structures). These findings highlight temperature as a critical hyperparameter for regulating LLM-optimizer symbiosis.

**Error rate study** As discussed in experiment 4.3, Table 6 quantifies the failure rates of LLM-generated objective functions across four scenarios and three prompt compositions, measured at the last iteration (10th iteration). For each scenario, LLM queries scale with time window duration: P50_C30_T300 (1 query per individual, 5 total responses for entire popsize) to P130_C80_T1200 (4 queries per individual, 20 total responses). Error rates are computed as $\frac{\text{\# invalid objectives}}{\text{total responses}}$.

When only $\mathcal{P}_{\text{sys}} \cup \mathcal{P}_{\text{geo}} \cup \mathcal{P}_{\text{data}}$ are provided, error rates reach 100% in all scenarios, as inadequate prompts yield semantically invalid formulations. Replacing $\mathcal{P}_{\text{data}}$ with $\mathcal{P}_{\text{model}}$ reduces errors to 20% (P50) and 36.7% (P130), demonstrating the necessity of task grounding via structural priors. Further

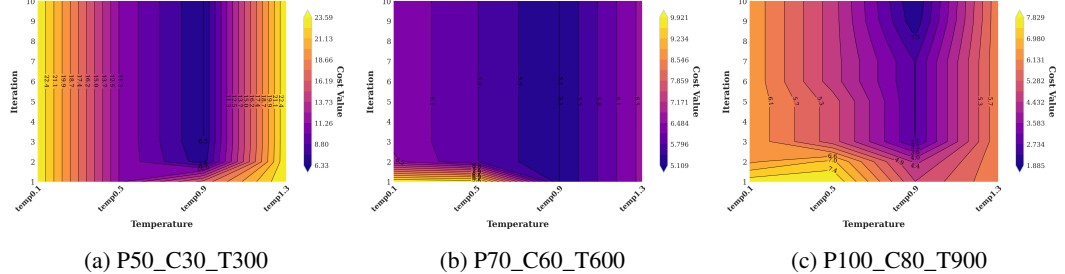

| (a) P50_C30_T300 | (b) P70_C60_T600 | (c) P100_C80_T900 |

Figure 12: Cost landscape across temperature settings and iterations.

incorporation of $\mathcal{P}_{\text{restriction}}$ minimizes errors to 0.0% in smaller scenarios (P50–P100) and 8.3% in P130, highlighting the role of constraint formalization. This indicates the critical role of progressive prompt enrichment, via model structuring and constraint formalization, in ensuring feasible and valid outputs from LLM-driven optimization pipelines. Additionally, for the latter two prompt compositions, error rates exhibit an upward trend with increasing scenario complexity (rising from 20.0% in P50 to 36.7% in P130 for $\mathcal{P}_{\text{model}}$, 0.0% in P50 to 8.3% in P130 for $\mathcal{P}_{\text{restriction}}$), reflecting the escalating fragility of LLM-generated objectives in high-dimensional spaces.

Table 6: Error rate analysis of LLM-generated solutions across prompt compositions

| Scenario | Prompt Composition | Run1 | Run2 | Run3 | Avg. |
|---|---|---|---|---|---|
| P50_C30_T300 | $\mathcal{P}_{\text{sys}} \cup \mathcal{P}_{\text{geo}} \cup \mathcal{P}_{\text{data}}$ | 100% | 100% | 100% | 100.0% |
| | $\mathcal{P}_{\text{sys}} \cup \mathcal{P}_{\text{geo}} \cup \mathcal{P}_{\text{model}}$ | 0% | 0% | 60% | 20.0% |
| | $+ \mathcal{P}_{\text{restriction}}$ | 0% | 0% | 0% | 0.0% |
| P70_C60_T600 | $\mathcal{P}_{\text{sys}} \cup \mathcal{P}_{\text{geo}} \cup \mathcal{P}_{\text{data}}$ | 90% | 100% | 100% | 96.7% |
| | $\mathcal{P}_{\text{sys}} \cup \mathcal{P}_{\text{geo}} \cup \mathcal{P}_{\text{model}}$ | 0% | 0% | 0% | 0.0% |
| | $+ \mathcal{P}_{\text{restriction}}$ | 0% | 0% | 0% | 0.0% |
| P100_C80_T900 | $\mathcal{P}_{\text{sys}} \cup \mathcal{P}_{\text{geo}} \cup \mathcal{P}_{\text{data}}$ | 100% | 100% | 100% | 100.0% |
| | $\mathcal{P}_{\text{sys}} \cup \mathcal{P}_{\text{geo}} \cup \mathcal{P}_{\text{model}}$ | 13.3% | 60% | 26.7% | 33.3% |
| | $+ \mathcal{P}_{\text{restriction}}$ | 0% | 0% | 0% | 0.0% |
| P130_C80_T1200 | $\mathcal{P}_{\text{sys}} \cup \mathcal{P}_{\text{geo}} \cup \mathcal{P}_{\text{data}}$ | 100% | 100% | 100% | 100.0% |
| | $\mathcal{P}_{\text{sys}} \cup \mathcal{P}_{\text{geo}} \cup \mathcal{P}_{\text{model}}$ | 55% | 25% | 30% | 36.7% |
| | $+ \mathcal{P}_{\text{restriction}}$ | 0% | 15% | 10% | 8.3% |

*Note:* Scenarios denote passenger counts (P), taxis (C), and time windows (T). Prompt components: system ($\mathcal{P}_{\text{sys}}$), geometry ($\mathcal{P}_{\text{geo}}$), argument/variable format($\mathcal{P}_{\text{data}}$), model blueprint ($\mathcal{P}_{\text{model}}$), and constraints ($\mathcal{P}_{\text{restriction}}$). $+\mathcal{P}_{\text{restriction}}$ denotes $\mathcal{P}_{\text{sys}} \cup \mathcal{P}_{\text{geo}} \cup \mathcal{P}_{\text{model}} \cup \mathcal{P}_{\text{restriction}}$.

### A.1.2 Future work

**Application layer scalability and simulation fidelity** Our experiments demonstrate effective optimization using a 32B-parameter LLM, suggesting smaller models can achieve comparable performance when properly constrained to domain-specific reasoning. In the future, we will integrate microscopic traffic simulators (e.g., SUMO [43]) to enable high-fidelity modeling of road network dynamics, including congestion, signal timing, and stochastic travel times with spatial correlations. This integration will support evaluation of our framework under realistic operational conditions, accounting for transient disruptions (e.g., accidents, road closures) that require dynamic re-planning.

**Reinforcement learning for LLM-Optimizer co-adaptation** The current framework treats the LLM as a fixed agent that iteratively adapts its objective-generation behavior through prompt-based interactions, it does not modify the model's internal parameters. However, the structure of our system, where the LLM proposes objectives and an external optimizer evaluates them, mirrors the actor-critic paradigm in reinforcement learning. This opens a promising direction: leveraging optimizer feedback (e.g., fitness costs) to directly fine-tune the LLM, potentially enabling efficient training with smaller

models. By integrating the optimizer as a critic that evaluates the LLM's proposed objectives, we can move beyond prompt engineering and into a setting where the LLM is explicitly trained to generate more effective objectives over time. Given previous success in using RL to effectively fine-tune LLMs [44], such optimizer-guided fine-tuning could enable more sample-efficient and domain-adaptive optimization pipelines, further bridging large language models and traditional solvers.

**Quantum-enhanced hierarchical optimization**  The proposed framework leverages LLM and classical optimization solvers in a hierarchical interaction to address sequential decision-making problems, we envision that advances in quantum computing could significantly enhance this paradigm. Although quantum computing is still in its early stages, it has shown strong potential in solving combinatorial and discrete optimization problems more efficiently than classical computers. As quantum hardware matures, it could accelerate both sides of the LLM-optimizer interaction, providing faster inference for LLM-guided heuristics and more efficient solutions from quantum-enhanced optimizers. This would enable tighter integration and more frequent interactions between the reasoning and solving components, opening the door to real-time implementations of our framework and its deployment in dynamic decision environments.

## A.2  Baseline Methods

**Manual-designed objectives**: We construct a set of handcrafted objective baselines by combining different dispatching priorities (e.g., Distance, Temporal, and Utilization) motivated by two prior works: [2], which proposes a distance-based assignment in modeling (used as our Distance-only baseline), and [24], which introduces MILP models incorporating temporal and utilization aspects (reflected in our Temporal × Utilization baseline). Building on these, we also include two additional combinations: Distance × Utilization, Distance × Temporal × Utilization. These rule-based objectives form the manual objectives baselines used to evaluate performance across various dispatching scenarios. The second-level routing adopts the optimization model from formulation 1.

**Default + RL-Seq**: As the name suggests, Default + RL-Seq uses the default objective Distance×Temporal ×Utilization for first-level assignment. However, for second-level routing, it utilizes the RL-based method proposed in [23] instead. More specifically, we trained a policy model to stochastically predict the order to pick up passengers grouped together after first-level assignment optimization, minimizing the total travel time. In our implementation, for both passengers and taxis, we use travel times to and from all possible locations as their input features to the policy model. And we use an architecture similar to [23] but disable glimpsing to enhance learning. During inference, a search-based strategy is used to decide the final pickup order.

**FullRL**: FullRL is an implementation of an online order dispatch policy [6], where decisions are made in a rolling horizon based on advantage function estimation $A_\pi(i,j) = \gamma^{\Delta_{t_j}} V(s'_{ij}) - V(s_i) + R_\gamma(j)$, where $\gamma$ is the discount factor, and $\Delta_{t_j}$ is the picking up time for the matched pair of driver $i$ and order $j$. The reward $R_\gamma(j)$ is the reward for dispatching a driver $i$ to order $j$. To sync with the problem settings in this paper, we modify the reward to reflect the waiting time of order $j$ instead of using the default value of 1. Consequently, the objective shifts from maximizing the service rate (as in the original paper) to minimizing total waiting times. The current state of a driver $i$ is represented as a two-dimensional vector (location and time). The value function of a driver's current state negatively impacts the advantage function ($-V(s_i)$). An action that assigns an order to a driver whose destination is in a more valuable region results in higher advantages, denoted as $V(S'_{ij})$. Finally, dispatching decisions at each timestamp are based on advantage function estimation.

**FunSearch**: FunSearch [15] utilizes LLM as well to produce methods for given problems. During testing, we made several adjustments to adapt to our taxi-passenger assignment problem. More specifically, expanding a generic prompt, we provided information on method input and output formats and more clearly defined the optimization purpose. Furthermore, we also adjusted the hyperparameters to encourage more explorations while searching for the final method. For each scenario, we sampled 20 methods before reporting the final results.

**EoH** EoH [16] is a framework that combines LLM and evolutionary computation to automate heuristic design. It utilizes five prompt-driven strategies and expresses heuristic ideas as natural language 'thoughts'. These thoughts are then converted by LLM into executable code, which helps improve and refine the original heuristics. In our experiments, we evaluate EoH on the taxi-passenger assignment problem using a constrained functional prompt. The goal is to minimize passenger waiting time while

ensuring balanced and efficient taxi utilization. We conduct evaluations across 9 scenarios, with 10 evolutionary iterations and a population size of 5.

### A.3 Dataset Description and Testing Scenario Generation

We use the New York taxi trip data from January to June 2024, publicly available from the NYC taxi and Limousine commission (`https://www.nyc.gov/site/tlc/about/tlc-trip-record-data.page`), and Chicago taxi trip data from January to June 2023, publicly available from Chicago data portal (`https://data.cityofchicago.org/Transportation/Taxi-Trips-2013-2023-/wrvz-psew/about_data`), to generate realistic simulation scenarios. For Manhattan, our analysis focuses on trips that originate and terminate within the downtown area. For Chicago, we focus on central district including *Loop* community (zone 32) and 18 neighboring communities. Rather than replaying historical trajectories, we extract key statistical properties from the dataset and use them to generate randomized scenarios.

Figure 13a illustrates the average spatial distribution of taxi trips over the six-month period in Downtown Manhattan area. Based on this distribution, we define peak hours as the time interval from 12:00 a.m. to 10:00 p.m. We further compute the empirical frequency of each OD pair, shown in Figure 13b. For each synthetic scenario, travel tasks are sampled according to the normalized OD frequency distribution, thereby preserving realistic spatial and temporal patterns observed in the real-world data.

In Figure 14a we show similar statistics of Chicago central district. The peaks hours in these areas are defined as from 8:00 a.m. to 7:00 p.m. Also, notice that taxi trips are highly concentrated on origin-destination pair (32, 8) and (32, 28), which is due to the fact that *Loop* (zone 32) is the central business district of Chicago and *Near North Side* (zone 8) and *Near West Side* (zone 28) are two regions next to *Loop* with large population.

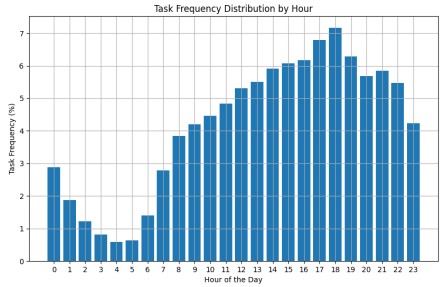

(a) Manhattan downtown average task distribution

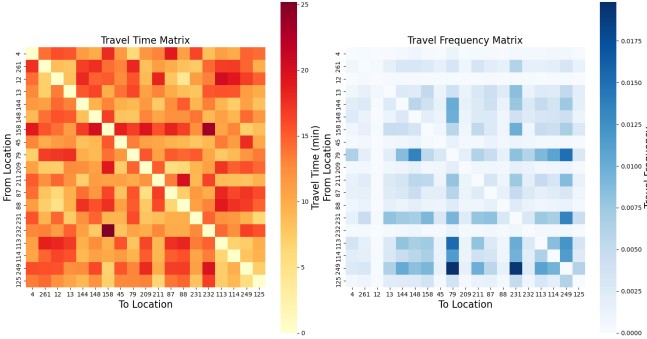

(b) Manhattan downtown travel time and frequency

Figure 13: Average task (passenger request) distribution and travel time and frequency statistics in Manhattan downtown

We construct two sets of nine synthetic simulation scenarios, derived separately from the statistical distributions of the New York TLC trip data and the Chicago taxi trip data. Each set consists of scenarios that share identical configurations in terms of passenger demand, taxi fleet size, and request

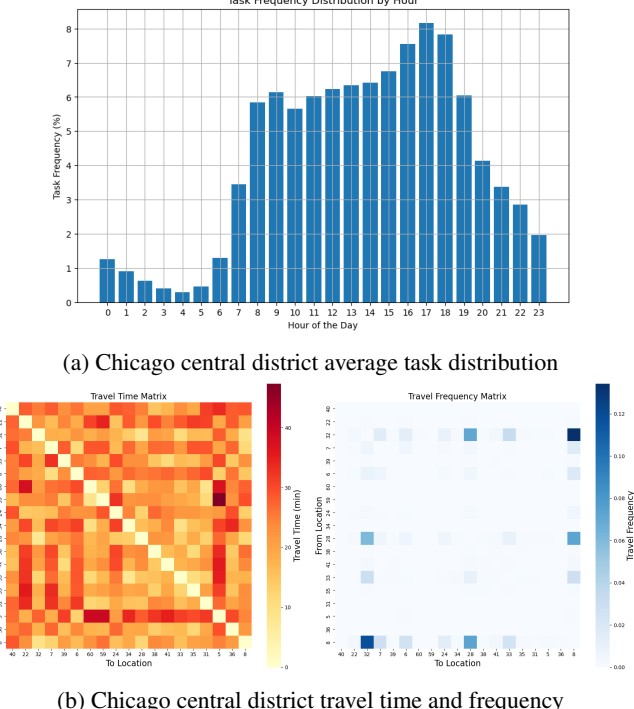

(a) Chicago central district average task distribution

(b) Chicago central district travel time and frequency

Figure 14: Average task (passenger request) distribution and travel time and frequency statistics in Chicago extended central district

time windows, but differ in spatial and temporal distribution characteristics specific to each city. Each scenario is parameterized by the following variables: (1) Passenger volume (P): Ranging from 35 to 200 requests, reflecting varying demand intensities. (2) Taxi fleet size (C): From 60 to 100 vehicles, testing scalability under resource constraints. (3) Request time window (T): 600s (10min), 900s (15min), or 1200s (20min), is the time window for distributing passenger request. Scenarios 1-9 in Table 2 and Table 3 corresponds to P35_C60_T600, P65_C80_T600, P100_C100_T600, P50_C60_T900, P100_C80_T900, P150_C100_T900, P70_C60_T1200, P130_C80_T1200, P200_-C100_T1200. For example, P200_C100_T1200 denotes a simulation scenario with 200 passenger requests occurring within a 20-minute time window, 100 available taxis in the system, and stochastic request generation based on the origin-destination distribution. The scenarios incrementally stress-test the system along these dimensions (Tables 2 and 3), avoiding historical replay while maintaining fidelity to real-world trip dynamics.

## A.4 Full Loop of Algorithm

The complete algorithmic process is outlined in Algorithm 2.

## A.5 Search Space of Dynamic Systems

A dynamic system in real-time decision-making (e.g., ride-hailing dispatching) is characterized by time-evolving states and sequential interdependent decisions. Unlike static OR problems (e.g., traveling salesman problem or bin packing), dynamic systems require continuous adaptation to changing environments, leading to exponentially growing search spaces. The problem at time $t$ can be formulated as: $\mathcal{S}_t = \{\mathcal{V}_t, \mathcal{P}_t, \mathcal{C}_t, \mathcal{D}_t\}$, where $\mathcal{S}_t$ is the system state at time $t$, is composed by the vehicle set $\mathcal{V}_t$ (taxis with locations and status), the passenger set $\mathcal{P}_t$ (pending passenger requests with pickup/dropoff locations in time windows), constraint set $\mathcal{C}_t$ (environment constraints) and historical decision trajectories up to $t$ - $\mathcal{D}_t$.

At each decision step $t$, the dispatcher selects an action $a_t \in \mathcal{A}_t$, where $\mathcal{A}_t$ represents the action space (e.g., assigning taxis to requests and visiting order of each taxi). The system transitions to a new state

**Algorithm 2** LLM-Optimizer Interaction Protocol

---

**Require:** Population size $N$, Control interval $\Delta t$, Scenario Information $\mathcal{U}$
**Ensure:** Evolved population of prompts and optimized dispatch strategies
 1: Initialize population $\mathcal{P}$ with $N$ individuals
 2: **for** generation = 1 **to** $max\_generations$ **do**
 3:     **for** round $i = 1$ **to** Popsize $N$ **do**                    ▷ Can trigger concurrently
 4:         parent response, evolution $way$ list = GETNEWINDIVIDUAL($\mathcal{P}$)     ▷ Current population
 5:         Initialize simulator with environment data $\mathcal{E}_0$
 6:         **for** step $t = 1$ **to** $T$ **do**                    ▷ Inner simulation loop
 7:             $\mathcal{E}_t \leftarrow$ GETSIMULATORSTATE()            ▷ Current vehicles, requests, etc.
 8:             **if** $way[t] =$ "way1" **then**                    ▷ Random generation
 9:                 $prompt_t \leftarrow$ GENERATERANDOMPROMPT()
10:             **else if** $way[t] =$ "way2" **then**                ▷ Memory consideration
11:                 $prompt_t \leftarrow$ GENERATEREVISEDPROMPT(parent response)
12:             **else**                                    ▷ ("way3")
13:                 $prompt_t \leftarrow$ GENERATEINNOVATEDPROMPT(parent response)
14:             **end if**
15:             $prompt_t \leftarrow prompt_t +$ INJECTCONTEXT($\mathcal{U}, \mathcal{E}_t$)     ▷ Add scenario and env data
16:             $response_t \leftarrow$ QUERYLLM($prompt_t$)                    ▷ Call API
17:             $objective \leftarrow$ PARSERESPONSE($response_t$)            ▷ Extract MILP objective
18:             Solve first-level assignment MILP:
19:                     min $objective$
20:                     $\mathbf{y}^* \leftarrow$ GUROBISOLVE(assign_model)
21:             Solve second-level sequencing MILP: ▷ Can trigger concurrently for different taxis
22:                     min $\sum_p max(0, \tau_p^{\text{pickup}} - \tau_p^{\text{req}})$
23:                     $\mathbf{x}^*, \tau^* \leftarrow$ GUROBISOLVE(sequence_model)
24:             $\mathcal{E}_{t+1} \leftarrow$ UPDATESIMULATOR($\mathbf{x}^*, \tau^*, \Delta t$)            ▷ Evolve $\Delta t$ seconds
25:         **end for**
26:         $fitness_i \leftarrow$ COMPUTEFITNESS($\mathcal{E}_1, \ldots, \mathcal{E}_T$)            ▷ Total system cost
27:         $\mathcal{P}_{new} \leftarrow \mathcal{P}_{new} + P_i$            ▷ $P_i = (\mathcal{E}_t, objective_t, fitness)_i$ for $t \in T$
28:     **end for**
29:     $\mathcal{P} \leftarrow$ SORTPOPULATION($\mathcal{P}, \mathcal{P}_{new}$)     ▷ Sort combined population and reserve the top $N$
30: **end for**

---

$\mathcal{S}_{t+1}$ based on the action $a_t$ is governed by: $\mathcal{S}_{t+1} = f(\mathcal{S}_t, a_t)$. The search space $\mathcal{S}$ for a dynamic system is the union of all possible state-action sequences over a finite horizon $T$: $\mathcal{S} = \bigcup_{t=0}^{T} \{(\mathcal{S}_t, a_t)\}$. The action space $\mathcal{A}_t$ at each step grows combinatorially with the number of vehicles $|\mathcal{V}_t|$ and the number of passenger requests $|\mathcal{P}_t|$.

- For the assignment between passengers and taxis at each step, its space can be $\mathcal{O}(|\mathcal{V}|^{|\mathcal{P}|})$.

- Assume $\mathcal{K}$ requests assigned to each taxi, the number of valid sequences is $\mathcal{K}!$, with $|\mathcal{V}|$ taxis, we can have $\mathcal{O}((\mathcal{K}!)^{|\mathcal{V}|})$.

- Given $T$ steps, the total search space grows as $\mathcal{O}((|\mathcal{V}|^{|\mathcal{P}|}\mathcal{K}!^{|\mathcal{V}|})^T)$.

This results in an extremely larger search space compared to traditional one-time OR problems, making it hard for direct resolution by LLM. While LLM continues to evolve, and current limitations may not persist in future iterations, the complexity of the search space necessitates a hybrid approach that integrates traditional optimization methods with LLM in a hierarchical framework. Therefore, we partition the problem into two levels:

**LLM-driven objective design in high-level assignment model**    LLM acts as meta-optimizer, dynamically proposing adaptive objectives to guide decision-making. The generated adaptive objective functions $\Phi_t^h$ encoding high-level goals, with current states $\mathcal{S}_t$ and historical information $H_{t-1}$ as inputs:

$$\min \Phi_t^h = LLM(\mathcal{S}_t, H_{t-1})$$
$$s.t. \sum_v y^{pv} = 1, \quad y^{pv} \in \{0, 1\}$$

On the one hand, the assignment layer's variables $y^{pv}$ lack direct visibility into sequencing outcomes (e.g., waiting times, route efficiency), on the other hand, a handcrafted objective (e.g., "minimize assignment distance") is myopic, as it ignores downstream sequencing impacts. To address this limitation, the LLM dynamically generates adaptive objectives $\Phi_t^{LLM}$ that implicitly steer assignments toward configurations favorable for low-level sequencing.

**Low-level sequencing model**  Given a fixed assignment plan, each taxi shall solve a routing problem to visit the assigned passenger one by one, aiming to minimize the total passenger waiting time:

$$\min \Phi_t^l(\mathcal{S}_t, \tau_t)$$
$$s.t. \ g(S_t, \tau_t, a_t^l) \leq 0$$

Where $\tau_t$ denotes the temporal variables (e.g., taxi arrival time), $a_t^l$ is the low-level action, namely, routing variables.

### A.6  Direct Modeling Approach

The complete taxi decision-making problem is formulated as a mixed-logic problem, where the objective is to minimize the passenger waiting time, incorporating constraints such as initial car and passenger location/time assignment, task connectivity constraint, vehicle travel dynamics and passenger boarding dynamics. The network we considered is defined as a directed graph $\mathcal{G} = \{\mathcal{S}, \mathcal{L}\}$, where $\mathcal{S}$ is the set of pick-up/dropoff stops in the network, with any stop $i \in \mathcal{S}$. $\mathcal{L}$ is the route path between any two stops. Multiple passengers (taxis) may share the same origins and/or destinations. we utilize the service point sets $\mathcal{P}$ ($\{(p, O^p)\}$, $\{(v, S^v)\}$, for origins) and $\mathcal{D}$ ($\{(p, D^p)\}$, $\{(v, E^v)\}$, for destinations) to distinguish individual tasks while preserving both passenger geographical information.

**Initial and final car location assignment**  Every time when a new planning occurs, the scheduler needs to know the initial taxi location, whether it is in idle state or driving on the road at this moment, we assume the time left for this taxi to arrive to its next immediate stop shall be known in advance, as depicted in constraints (3a) and (3b), respectively. Also, the taxi shall finally go back to the sink location after serving all assigned passengers, as captured in constraint (3c).

$$\sum_{i \in \mathcal{P}} x_{(v,S^v),i}^v = 1 \tag{3a}$$

$$AR_{(v,S_v)}^v = \tilde{t}^v \tag{3b}$$

$$\sum_{i \in \mathcal{P}} x_{i,(v,E^v)}^v = 1 \tag{3c}$$

where $x_{ij}^v$ indicates whether pick-up & dropoff pair $ij$ is selected for taxi $v$. $AR_i^v$ is the arrival time of car $v$ for pick-up service $i$, where $i \in \mathcal{P}$.

**Passenger assignment constraints**  Each passenger can at most board on one car, as illustrated in (4).

$$\sum_v y^{pv} = 1 \tag{4}$$

Also, whenever a taxi $v$ is assigned to serve passenger $p$, it will firstly reach stop $O^p$ to pick up the passenger, and subsequently proceed towards stop $D^p$ to deliver the passenger to its destination. Thus, the operating taxi $v$ shall confirm the edge OD for the selected passenger $p$.

$$x_{(p,O^p),(p,D^p)}^v \geq y^{pv} \tag{5}$$

**Service connectivity constraints**   Whenever a group of services, each service incorporates a pick-up point and a dropoff point, are assigned to the taxi, for any two services, the taxi must complete one service before starting the next. In other words, if passengers $p$ and $p'$ are both assigned to the same taxi $v$, then either passenger $p$ is served first or the passenger $p'$ is served first. This order is reflected as a conservation law of typical Vehicle routing problem in current problem formulation, as captured in constraint (6).

$$\forall v \in V, \forall i \in \mathcal{P}, \forall j \in \mathcal{D}$$

$$\sum_{i \in \mathcal{P}} x_{ij}^v = \sum_{q \in \mathcal{P}} x_{jq}^v \tag{6a}$$

$$\sum_{j \in \mathcal{D}} x_{ji}^v = \sum_{k \in \mathcal{D}} x_{ik}^v \tag{6b}$$

Also, each passenger must be served by one taxi, as illustrated in (4), accordingly, each pick-up or dropoff service point can only be visited one time by any of the taxi, as described below:

$$\forall v \in V, \forall i \in \mathcal{P}, \forall j \in \mathcal{D}$$

$$\sum_v \sum_{i \in \mathcal{P}} x_{ij}^v = 1 \tag{7a}$$

$$\sum_v \sum_{j \in \mathcal{D}} x_{ij}^v = 1 \tag{7b}$$

**Arrival and departure time constraints**   If the route edge OD for passenger $p$ is selected for taxi $v$, then arrival time at dropoff point is equal to the sum of the pick up time at origin point and the OD travel time. Also, departure time at dropoff point is larger than its arrival time.

$$x_{(p,O^p),(p,D^p)}^v = 1 \rightarrow$$
$$AR_{(p,D^p)}^v = DP_{(p,O^p)}^v + TR_{O^p D^p} \tag{8a}$$
$$DP_{(p,D^p)}^v \geq AR_{(p,D^p)}^v \tag{8b}$$

where $DP_j^v$ is departure time of car $v$ for dropoff service $j$, where $j \in \mathcal{D}$. $A \rightarrow B$ is a logic constraint, indicating if $A$ is true, then implies $B$.

Also, if passenger $p'$ is served immediately after passenger $p$ for taxi $v$, then arrival time at pick-up point for passenger $p'$ is equal to the sum of the departure time of passenger $p$ at its dropoff point and the edge travel time. Also, departure time at pick-up point shall equal to the maximum value between the taxi arrival time and the passenger arrival/request time.

$$x_{(p,E^p),(p',O^{p'})}^v = 1 \rightarrow$$
$$AR_{(p',O^{p'})}^v = DP_{(p,D^p)}^v + TR_{D^p O^{p'}} \tag{9a}$$
$$DP_{(p',O^{p'})}^v = \max(AR_{(p',O^{p'})}^v, T^{p'}) \tag{9b}$$

**Objective function**   The goal is to minimize the total passenger waiting time, as captured below:

$$J = \sum_p \sum_v WT^{pv} \tag{10}$$

where waiting time $WT^{pv}$ is obtained from constraints below, only if passenger $p$ is assigned to taxi $v$, corresponding waiting time $WT^{pv}$ shall exist, otherwise, should be 0:

$$y^{pv} = 1 \rightarrow WT^{pv} = DP_{(p,O^p)}^v - T^p \tag{11a}$$
$$y^{pv} = 0 \rightarrow WT^{pv} = 0 \tag{11b}$$

While alternative objectives, such as platform profit or driver earnings, can be incorporated to account for broader stakeholder impacts, this work does not aim to design a comprehensive or multi-faceted objective. Instead, it focuses on the interaction between the LLM and the optimizer, demonstrating how high-level goals specified via natural language can evolve into low-level objectives that guide decision-making. Extensions to other objectives are straightforward and left for future work.

The proposed models above is formulated as a mixed logical model, incorporating both linear constraints and logical constraints (8) (9), and (11). These logical constraints can be equivalently transformed into linear constraints using the big-M method [45], with detailed formulations provided in Appendix A.7. Although the problem can be reformulated as a MILP model and solved directly using commercial solvers such as Gurobi to obtain optimal results, its large scale presents significant computational challenges. For instance, in a taxi-hailing system, each time can trigger a scenario with 80 taxis and 50 passenger requests, the decision variables $x_{ij}^v$ alone result in 200,000 (80*50*50) variables in MILP. Given the short decision-making intervals required in real-time operations, solving such a large-scale MILP optimally within the available time frame is impractical. Therefore, the problem is typically formulated as a sequential approach: first, solving an assignment problem to allocate a subset of passengers to a group of taxi drivers, followed by solving a traveling salesman problem with additional time constraints for each taxi independently in parallel.

### A.7 Logic Constraints Conversion

In Section A.6, the problem has been formulated as a mixed logical dynamic model, which involves logic constraints, e.g.,arrival and departure time constraints (8), (9), and waiting time constraints (11). All these logic constraints can be converted to linear constraints via big-M method as follows.

**Proposition 1** *Replacing logic constraints (8) with Inequalities (12) in the model leads to the same solution.*

**Proof:** Let $M_1$ be sufficiently large, and satisfies $M_1 \geq max\{\pm(AR_{(p,D^p)}^v - DP_{(p,O^p)}^v - TR_{(O^p,D^p)}), \pm(AR_{(p,D^p)}^v - DP_{(p,D^p)}^v)\}$ for $v \in V, p \in P\}$, then constraint (8) can be rewritten as:

$$\forall v \in V, p \in P$$
$$AR_{(p,D^p)}^v - DP_{(p,O^p)}^v - TR_{(O^p,D^p)} \leq M_1(1 - x_{(p,O^p)(p,D^p)}^v) \tag{12a}$$
$$- AR_{(p,D^p)}^v + DP_{(p,O^p)}^v + TR_{(O^p,D^p)} \leq M_1(1 - x_{(p,O^p)(p,D^p)}^v) \tag{12b}$$
$$AR_{(p,D^p)}^v - DP_{(p,D^p)}^v \leq M_1(1 - x_{(p,O^p)(p,D^p)}^v) \tag{12c}$$
$$- AR_{(p,D^p)}^v + DP_{(p,D^p)}^v \leq M_1(1 - x_{(p,O^p)(p,D^p)}^v) \tag{12d}$$

**Proposition 2** *Replacing logic constraints (9) with Inequalities (13) in the model leads to the same solution.*

**Proof:** Let $M_2$ be sufficiently large, and satisfies $M_2 \geq max\{\pm(AR_{(p',O^{p'})}^v - DP_{(p,D^p)}^v - TR_{(D^p,O^{p'})}), AR_{(p',O^{p'})}^v - DP_{(p',D^{p'})}^v, T^{p'} - DP_{(p',D^{p'})}^v\}$ for $v \in V, p \in P\}$, then constraint (8) can be rewritten as:

$$\forall v \in V, p \in P$$
$$AR_{(p',O^{p'})}^v - DP_{(p,D^p)}^v - TR_{(D^p,O^{p'})} \leq M_2(1 - x_{(p,E^p)(p',D^{p'})}^v) \tag{13a}$$
$$- AR_{(p',O^{p'})}^v + DP_{(p,D^p)}^v + TR_{(D^p,O^{p'})} \leq M_2(1 - x_{(p,E^p)(p',D^{p'})}^v) \tag{13b}$$
$$DP_{(p',D^{p'})}^v - AR_{(p',O^{p'})}^v \geq -M_2(1 - x_{(p,E^p)(p',D^{p'})}^v) \tag{13c}$$
$$DP_{(p',D^{p'})}^v - T^{p'} \geq -M_2(1 - x_{(p,E^p)(p',D^{p'})}^v) \tag{13d}$$

**Proposition 3** *Replacing logic constraints (11) with Inequalities (14) in the model leads to the same solution.*

**Proof:** Let $M_3$ be sufficiently large, and satisfies $M_3 \geq max\{\pm(WT^{pv} - DP_{(p,O^p)}^v - T^p), \pm(WT^{pv})\}$ for $v \in V, p \in P\}$, then constraint (11) can be rewritten as:

$$\forall v \in V, p \in P$$
$$WT^{pv} - DP_{(p,O^p)}^v - T^p \leq M_3(1 - y^{pv}) \tag{14a}$$
$$- WT^{pv} + DP_{(p,O^p)}^v + T^p \leq M_3(1 - y^{pv}) \tag{14b}$$
$$WT^{pv} \leq M_3 y^{pv} \tag{14c}$$
$$WT^{pv} \geq -M_3 y^{pv} \tag{14d}$$

## A.8  Dynamic Simulator Environment Setup

In the simulator, the state of the traffic network at time instant $k$ is simplified as $y(k) = \{d_i(k), t_i^{\mathrm{arr}}(k), i \in \mathcal{I}\}$, where $i$ is the index of the taxi, $\mathcal{I}$ is the set of all taxis, $d_i(k)$ is the location the $i$-th taxi at time instant $k$ is travelling to, and $t_i^{\mathrm{arr}}(k)$ is the time of the $i$-th taxi arriving that location. The following conventions are made for state and task update in the simulation:

**1)**: the task being executed will not be reassigned.

**2)**: if taxi has no task assigned, it will stay at destination of the last task.

The command issued from optimizer is a set of task sequences for each taxi in the traffic system. Each task has four parameters: origin $o$, destination $d$, passenger arrival time $t^{\mathrm{start}}$, taxi departure time $t^{\mathrm{dep}}$. We denote the command for the $i$-th taxi from the optimizer at time instant $k$ as $u_i^{opt}(k)$ and the command being executed as $u_i(k)$. When the command from optimizer is issued to taxis, the command $u_i(k)$ will be set as $u_i^{opt}(k)$. The command $u_i(k)$ is defined as $u_i(k) = \{\mathrm{task}_{i,0}(k), \ldots, \mathrm{task}_{i,N_i}(k)\}$, where $\mathrm{task}_{i,j}(k) = (o_{i,j}(k), d_{i,j}(k), t_{i,j}^{\mathrm{arr}}(k), t_{i,j}^{\mathrm{start}}(k), t_{i,j}^{\mathrm{dep}}(k))$ collecting origin, destination, taxi arrival time, passenger arrival time and taxi departure time from task origin, respectively. With system state $y(k)$ and command $u(k) = \{u_i(k), i \in \mathcal{I}\}$, the state is updated as $y(k+1) = f(y(k), u(k), k)$ where $f(\cdot, \cdot, \cdot)$ is given as in Algorithm 3 for each $k$. When receiving new computed $u^{opt}(k) = \{u_i^{opt}(k), i \in \mathcal{I}\}$, the state and command shall be updated as in 4.

---

**Algorithm 3** State Transition Model

---

1: **Input**: Current state $y(k)$, commands $u(k)$
2: **Output**: Next state $y(k+1)$
3: $k \leftarrow k + 1$
4: **for** each taxi $i \in \mathcal{I}$ **do**
5:     **if** $u_i(k) \neq \emptyset$ **then**
6:         $(o_c, d_c, t_c^{\mathrm{arr}}, t_c^{\mathrm{start}}, t_c^{\mathrm{dep}}) \leftarrow \mathrm{head}(u_i(k))$
7:         **if** $k = t_i^{\mathrm{arr}}$ **then**
8:             **if** $d_i = o_c$ **then**
9:                 Update destination: $d_i \leftarrow d_c$
10:                 Update arrival time: $t_i^{\mathrm{arr}} \leftarrow t_c^{\mathrm{dep}} + TR_{o_c, d_c}$
11:             **else**
12:                 $\mathrm{pop}(u_i(k))$
13:                 **if** $u_i(k) \neq \emptyset$ **then**
14:                     $(o_c, d_c, t_c^{\mathrm{arr}}, t_c^{\mathrm{start}}, t_c^{\mathrm{dep}}) \leftarrow \mathrm{head}(u_i(k))$
15:                     Move to next task: $t_i^{\mathrm{arr}} \leftarrow \max\{k, t_c^{\mathrm{start}}\} + TR_{d_i, o_c}, d_i \leftarrow o_c$
16:                 **end if**
17:             **end if**
18:         **end if**
19:     **else**
20:         Mark taxi $i$ as idle
21:     **end if**
22: **end for**

---

**Algorithm 4** Command Update Protocol

---

1: **Input**: Optimized commands $u^{\mathrm{opt}}(k)$, current state $y(k)$
2: **Output**: Updated commands $u(k+1)$
3: **for** each taxi $i \in \mathcal{I}$ **do**
4:     **if** $u_i(k) = \emptyset$ **then**
5:         $u_i(k+1) \leftarrow u_i^{\mathrm{opt}}(k)$
6:     **else**
7:         Merge commands: $u_i(k+1) \leftarrow \mathrm{append}(u_i(k), u_i^{\mathrm{opt}}(k))$
8:     **end if**
9: **end for**

---

## A.9 Prompt Engineering

In this section, we provide the details of prompt strategies used in our method.

### A.9.1 Scenario prompt compositions

Figure 15 illustrates the scenario prompt structure introduced in section 3.2.1, where the scenario prompt is composed by $\mathcal{P}_{\text{sys}}$, $\mathcal{P}_{\text{geo}}$ and $\mathcal{P}_{\text{model}}$, respectively.

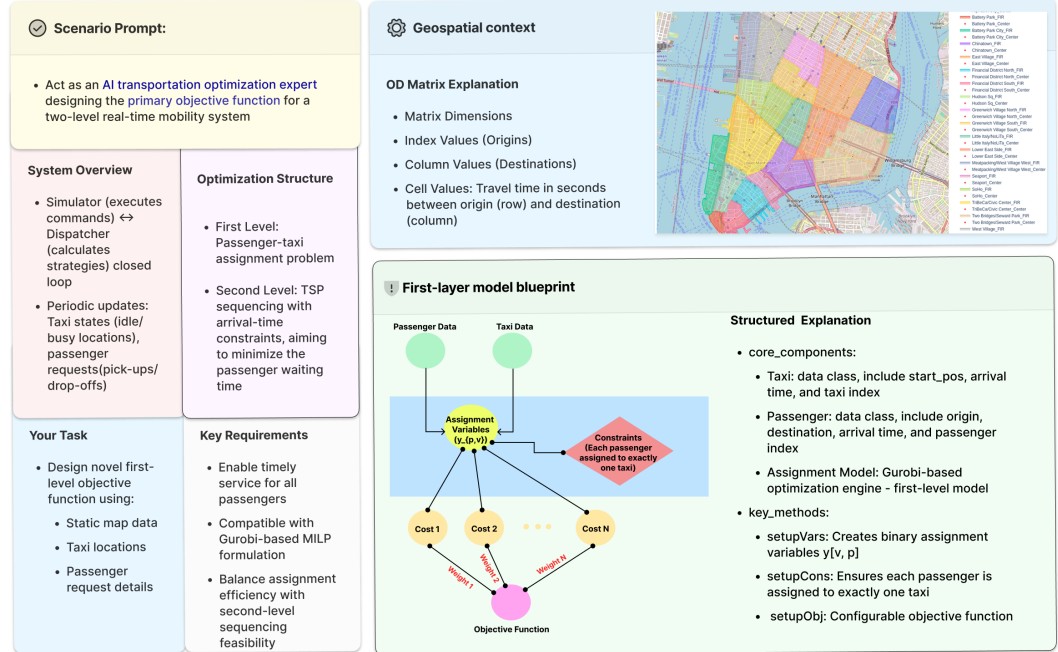

Figure 15: Scenario prompt structure provided to LLM, including system architecture and role defintion (left panel), geospatical content (upper right panel), first-layer model blueprint (lower right panel)

**System architecture & role definition ($\mathcal{P}_{\text{sys}}$)**  The system architecture and role definition establish the LLM as an adaptive objective designer rather than an end-to-end solver.

*System Architecture*: We provide a brief overview of the system architecture, including the communication between the simulator and the dispatcher, the two-level decomposition of the dispatching problem, namely, $\mathcal{M}_{\text{assign}} : \min_y \mathcal{L}_{\text{assign}}(y|\theta_{\text{LLM}})$ and $\mathcal{M}_{\text{route}} : \min_x \mathcal{L}_{\text{route}}(x|y^*)$.

*Role Specification*: Define the LLM's role to constrain its outputs to objective function components compatible with the Gurobi [42] solver API using template-based generation.

**Geospatial context $\mathcal{P}_{\text{geo}}$**  As the problem is related to ride-hailing services, incorporating geospatial context is essential. Therefore, we provide the testing environment details to the LLM. Utilizing the publicly available New York or Chicago taxi dataset, our analysis focuses on 19 zones in either Downtown Manhattan or extended Central Chicago. Static map data is represented as an OD matrix $W_{OD}$, encoding spatial semantics via the Manhattan zone graph $G = (\mathcal{V}, \mathcal{E}, W_{OD})$ where $W_{\text{OD}}[i,j] = t_{ij} \in \mathbb{R}^+, \quad \forall (v_i, v_j) \in \mathcal{E}$.

**First-layer model blueprint $\mathcal{P}_{\text{model}}$**  The first-level assignment model class structure is exposed to guide LLM-compatible objective design. This class structure provides a framework for the LLM to work within, ensuring that its output is consistent with the overall model design. By having access to the class structure, the LLM can understand the relationships between different variables and design an objective function that takes these relationships into account.

### A.9.2 Harmony search prompt instruction

As discussed in section 3.2.3, our evolutionary operators employ structured instruction blocks to guide LLM-based prompt optimization. Specific instructions are listed as follows:

**Heuristic Improvement** ($W_2$) **Instruction Block:**   Focuses on incremental objective refinement:

    (a) Temporal alignment: Incorporate taxi-passenger arrival time coordination.

    (b) Resource weighting: Adaptive taxi utilization coefficients

    (c) Structural preservation: Maintain 50% legacy objective components.

**Innovative Generation** ($W_3$) **Instruction Block:**   Promotes paradigm-level innovations:

    (a) Emphasis on goals: design first-level decisions $y_{v,p}$ minimize *expected* second-level waiting:

$$\min \sum_{p \in \mathcal{P}} \mathbb{E} \left[ \max \left( DP_p^{(2\mathrm{nd})} - t_p^{\mathrm{arr}}, 0 \right) | y_{v,p} \right]$$

    (b) Multi-horizon optimization: Joint current/future state consideration

    (c) Dynamic weight adaptation: Time-varying priority coefficients

### A.9.3 Gurobi format restriction

To ensure the objective function is compatible with Gurobi, additional constraints are introduced to enhance its feasibility. Two types of restrictions are imposed: 1. *Admissible Terms*: These constraints define the complete structure of the objective function, ensuring all components in the function adhere to the class requirements. 2. *Expression Construction Rules*: Guidelines specifying the valid operations and transformations permitted in Gurobi's expression framework to maintain solver compatibility.

**Admissible components**   The generated objective functions must utilize only the following class elements, which are also summarized in Table 7:

- **Decision variables**: Binary assignments: `self.y[v,p]` $\in \{0, 1\}$
- **Static parameters**:
  Distance matrix: `self.distMatrix[o][d]`
  Big-M constant: `self.M`
- **Taxi state** (read-only):
  Current position: `self.taxi[v].start_pos`
  Availability time: `self.taxi[v].arrival_time`
- **Passenger state** (read-only):
  Origin/Destination: `self.passenger[p].origin`, `self.passenger[p].destination`
  Request time: `self.passenger[p].arrTime`

Table 7: Model specification mapping

| Concept | Mathematical Form | Code Implementation |
|---|---|---|
| Assignment variable | $y_{v,p}$ | `self.y[v,p]` |
| Travel time matrix | $TR_{o,d}$ | `self.distMatrix` |
| Taxi availability | $\tau_v^{\mathrm{avail}}$ | `self.taxi[v].arrival_time` |
| Passenger request time | $\tau_p^{\mathrm{req}}$ | `self.passenger[p].arrTime` |
| Weighted sum | $\sum w_i c_i$ | `sum(w * c for w, c in zip(weights, costs))` |

**Expression construction rules**    All objectives must adhere to:

1. **Structural requirements**:
   Costs stored in list `costs` (1-5 elements)
   Weights in list `weights` (length matching `costs`)
   Final objective: `sum(w*c for w,c in zip(weights, costs)`

2. **Gurobi expression rules**:
   - Use **`gb.quicksum()/gb.max_()/gb.abs_()`** only when containing variables
   - Use **`sum()/max()/abs()`** when working with parameters
   - Quadratic terms via variable multiplication (`y[v,p]*y[v,q]`)
   - Never multiply Gurobi variables with Gurobi expressions
     - `Invalid:  y[v,p] * gb.max_(expression_with_vars, 0)`
   - Never use Python if/else with Gurobi variables/expressions

**Generation protocol**    The following template is provided to LLM for valid objective construction:

```python
def dynamic_obj_func(self):
    cost1 = [Proper Gurobi expression]
    cost2 = [Proper Gurobi expression]
    # Add more components as needed

    # Custom weights (match costs length)
    weights = [
        [Your custom weight 1],
        [Your custom weight 2],
        # Add matching weights
        ]
    objective = sum(w*c for w,c in zip(weights, costs))
    self.model.setObjective(objective, gb.GRB.MINIMIZE)
```

### A.9.4   LLM response examples

Figure 17 and Figure 18 illustrate four LLM response examples generated by the proposed method. The first-layer model blueprint defines the default objective function, serving as a foundational heuristic for the LLM, as captured in Figure 16. The listed four objective function variants capture different optimization strategies:

Figure 16: Default objective code used in first-layer model blueprint

**Weighted multi-objective with dynamic penalties**    As captured in left panel of Figure 17, unlike the static weights in the default, this variant introduces dynamic penalties (e.g., time-scaled idle taxi prioritization in $cost_2$, future availability adjustments in $cost_5$) and amplifies priority weights (e.g., 80 for waiting time, 1000 for load imbalance). By dynamically scaling penalties based on real-time conditions (e.g., taxi arrival vs. passenger request times), it adaptively balances short-term efficiency and long-term fleet availability, addressing the default's rigidity in handling time-dependent scenarios.

**Quadratic load balancing with reassignment penalty**    While the default penalizes load imbalance via $cost_4$, this variant (shown in right panel Figure 17) explicitly enforces fairness through a squared deviation from the average passenger-per-taxi ratio (load_balance). Furthermore, this variant incorporates a quadratic reassignment penalty ($cost_3$) to minimize frequent taxi reassignments. This dual focus effectively reduces reassignment churn and strengthens fairness, addressing the limitations of the default model in enforcing balanced allocations.

**Busy taxi penalty with load balancing**    Replacing the default's simplistic waiting time penalty, this variant (captured in left panel of Figure 18) directly penalizes assignments to busy taxis using a > operator ($cost_4$) to check if a taxi's availability precedes the passenger's request. This variant incorporates the full trip duration into $cost_2$, extending beyond the default model's focus on pickup times alone. Additionally, it maintains quadratic load balancing to distribute demand more effectively. By considering both real-time availability and total trip effort, this approach prevents excessive assignments to already occupied taxis, addressing a key limitation of the default model.

**Sequential assignment with dropoff-pickup chaining**    Unlike the default's isolated trip modeling, this variant (captured in right panel of Figure 18) introduces a sequential chaining mechanism in $cost_3$, penalizing dropoff-to-pickup travel time between passengers ($y[v, p] * y[v, p\_]$). It also adds bidirectional waiting time penalties ($cost_1$ for taxi delay, $cost_5$ for passenger earliness), whereas the default only penalizes taxi delays. By optimizing trip sequences and accounting for both waiting constraints, it enhances fleet utilization efficiency, which is an improvement over the default's single-trip optimization approach.

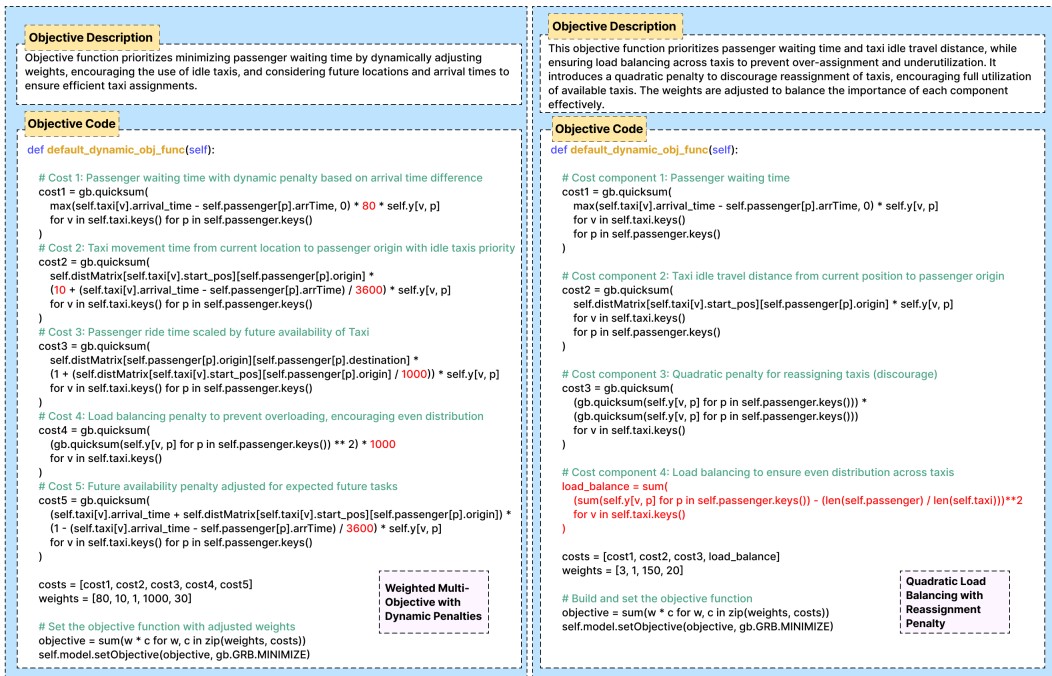

Figure 17: LLM response examples, including dynamic penalty variant (left panel) and quadratic loading balancing variant (right panel)

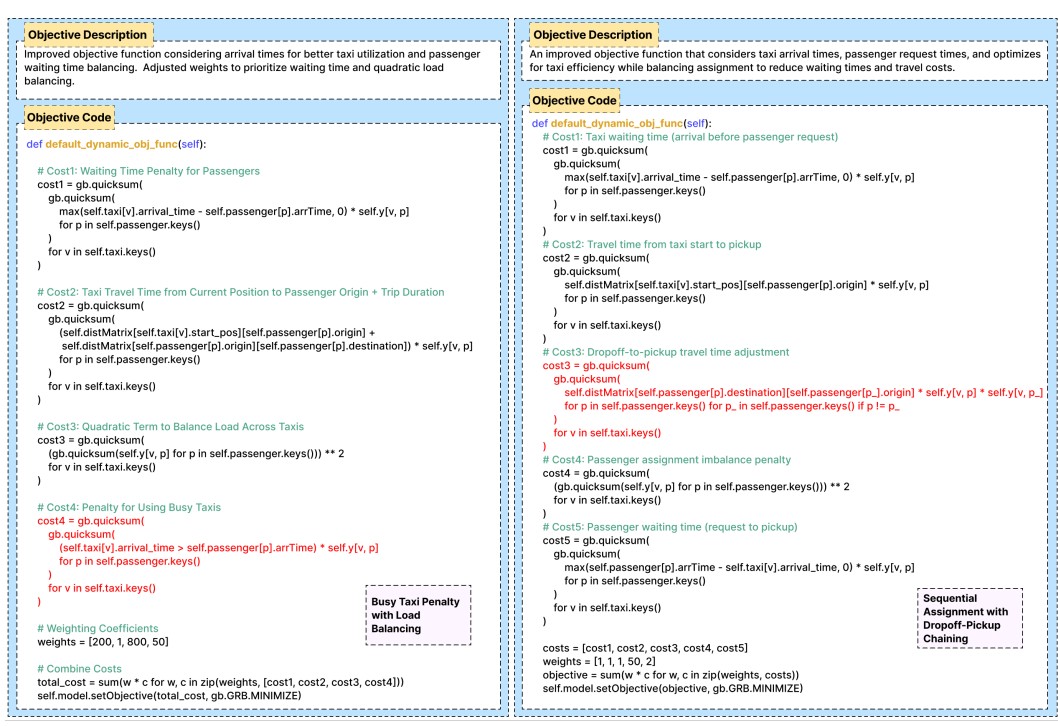

Figure 18: LLM response examples, including busy taxi variant (left panel) and sequential assignment variant (right panel)