# OpenReview forum: "Hierarchical Optimization via LLM-Guided Objective Evolution for Mobility-on-Demand Systems"
_NeurIPS.cc/2025/Conference — NeurIPS 2025 poster_

### Official Review · Reviewer_8CYJ · 2025-06-26

**Clarity:** 3
**Significance:** 2
**Originality:** 2
**Rating:** 4
**Confidence:** 3

**Summary:**

This paper proposes a hybrid method that integrates LLMs with exact solvers to address dynamic sequential decision-making in mobility-on-demand systems (which has an extremely high-dimensional search space and thus necessitates decomposition into high- and low-level optimization problems). The LLM in the method acts as a meta-optimizer that dynamically formulates high-level objectives via prompt-based evolution (employing the HS algorithm for the evolutionary process). The exact solver then addresses the optimization problems established at the corresponding time step. The method decomposes the ride-hailing problem hierarchically into assignment and routing subproblems, with LLMs bridging the cognitive gap between high and low levels by generating semantically-rich objectives that implicitly encode low-level routing dynamics.

**Questions:**

NA

**Ethical Concerns:**

["NO or VERY MINOR ethics concerns only"]

**Final Justification:**

Thank you for your response. Most of my concerns have now been satisfactorily addressed, so I have increased my rating. I appreciate the effort involved in preparing additional results during the rebuttal period. However, I strongly recommend that the author(s) include a more comprehensive and robust empirical study in the camera-ready version (if accepted), with additional experiments to demonstrate how the new method improves performance. I believe these results will interest a broad audience.

**Quality:**

3

**Strengths And Weaknesses:**

This paper addresses a practical problem, which I consider highly valuable. On the one hand, it proposes a novel problem formulation for ride-hailing systems and provides a formal analysis of the associated search space complexity. On the other hand, this work is the first to integrate LLMs with exact solvers for sequential dynamic decision-making systems. The proposed method is training-free and overcomes the data inefficiency challenges inherent in reinforcement learning, as well as the rigidity of manually designed objectives in traditional decomposed optimization. A key feature is its ability to dynamically adjust the high-level optimization problem according to $\mathcal{S}\_t$ and $H\_{t-1}$. The experimental results are also promising.

My concerns are as follows:
- What is the rationale for using the HS algorithm? Can it be replaced by other EAs?
- What are the specific objects designed by FunSearch and EoH in the experiments? Can the HS algorithm be substituted with FunSearch or EoH? I think that these two methods essentially also design functions.
- The ablation study on the operators of the HS algorithm is missing.
- Due to the use of LLMs, the worst-case runtime of the proposed method is difficult to predict.
- I think the author(s) should clarify that all information contained in $\mathcal{S}\_t$ and $H\_{t-1}$ positively contributes to the method's performance. Besides, could author(s) provide a specific example of $\mathcal{S}\_t$ and $H\_{t-1}$?
- Is it possible to dynamically adjust the parameters $\alpha, \beta, \gamma$ to improve the performance of manual objectives? Moreover, the experiments do not specify the settings of these parameters.
- Can multiple high-level objectives be pre-designed using LLMs and the appropriate objective be automatically selected at each time step? I think this could significantly reduce both runtime and the cost associated with using LLMs.

---

> ### Author Rebuttal · Authors · 2025-07-30
>
> **We appreciate the reviewers’ thorough feedback. We have grouped the identified weaknesses and concerns into the following comments for clarity and response.**
>
> **Comment 1**. What is the rationale for using the HS algorithm? Can it be replaced by other EAs?
>
> **Response**: Thanks for the reviewer's comments. We select HS because it can address the key challenge for evolving LLM prompts: encoding complex problem context, such as environment states and past objectives, into natural language in a clear and effective way. Traditional evolutionary strategies like Genetic Algorithms (GA), which rely on **crossover** and **mutation**, often produce semantically inconsistent text, **disrupting the alignment between prompt structure and evaluation scores**, making fitness unreliable. Furthermore, they **require complex parsing to maintain coherence**, ultimately reducing robustness and effectiveness in prompt optimization. HS avoids these issues by using two mechanisms: **Memory Consideration (HMCR)**: Copies full parent prompts, preserving structure and associated fitness. **Pitch Adjustment (PAR)**: Applies minimal edits, enabling controlled refinement without breaking grammar or semantics.
>
> **Comment 2**. What are the specific objects designed by FunSearch and EoH? Can HS be substituted with FunSearch or EoH? I think that these two methods essentially also design functions.
>
> **Response**: Thanks for the reviewer's comments. While FunSearch and EoH indeed involve function design, their formulation and target applications differ fundamentally from ours in several key aspects:
> - **Problem Nature**: FunSearch and EoH target *static, single-agent* combinatorial problems such as TSP and bin packing, where solutions can be constructed greedily without temporal coupling. In contrast, our task involves *multi-agent, dynamic decision-making* under time-dependent constraints (see Eqs. (1g)–(1m) in the paper).
> - **Output Type**: FunSearch/EoH prompt LLMs to directly generate *executable code function*, e.g., `def next_location(unvisited, current_pos)`. Our framework instead prompts the LLM to generate *mathematical objective functions*, which are optimized by a mixed-integer linear programming (MILP) solver to produce globally feasible, coordinated assignments.
> - **Conformity to System Constraints**: FunSearch/EoH bypass optimization solvers and cannot enforce hard constraints. More strict prompt design is required to avoid generating infeasible solutions in multi-constraint settings. However, FunSearch and EoH cannot guarantee to generate feasible solutions when the problem constraints are complex. On the other hand, HS in our method evolves high-level objective structures that remain compatible with MILP formulations (see paper Appendix A.9.2–A.9.3 for Gurobi-specific prompt constraints). This integration of semantic generation (LLM) and numerical grounding (solver) is central to our approach.
>
> Thus, FunSearch and EoH cannot be used to substitute HS. Doing so will lead to a lack of system-wide interactions, as FunSearch and EoH mostly generate local, next-visit decisions without global coordination. Furthermore, there will be more infeasible solutions, as essential hard system constraints cannot be enforced.
>
> **Comment 3**. The ablation study on the operators of the HS algorithm is missing.
>
> **Response**: Thanks for the reviewer's comments. Due to the main paper’s strict page limit, the HS parameter study was included in Appendix A.1.1 under the title 'Evolutionary hyperparameter study' (line 604). We can relocate this section to the main body of the paper accordingly. In this Appendix subsection, we describe how the parameters HMCR and PAR are adjusted to control the selection frequency of each operator.
>
> **Comment 4**. Due to the use of LLMs, the worst-case runtime of the proposed method is difficult to predict.
>
> **Response**: Thanks for the reviewer's comments. We would like to clarify that this paper focuses on the **conceptual and algorithmic design of a novel LLM-guided optimization framework**. In practical deployments, especially for latency-sensitive applications like transportation, LLMs can be deployed locally to minimize network delays and ensure tighter runtime guarantees.
> ```
> To understand our problem, we also provide the run time in Tables I and II below for our experiments. Although runtime grows with simulation scale, it remains significantly below the actual simulation duration. For example, Case 7 in Table I (20min simulation) completes in ~4 minutes, despite multiple optimization rounds, confirming the framework's practical runtime efficiency.
> ```
> Table I. Run time (s) for Manhattan cases
> | Sim Duration (s) | 600 | 600 | 600 | 900 | 900 | 900 | 1200 | 1200 | 1200 |
> |--------|--------|--------|--------|--------|--------|--------|--------|--------|--------|
> | Cases | Case 1 | Case 2 | Case 3 | Case 4 | Case 5 | Case 6 | Case 7 | Case 8 | Case 9 |
> | Run Time (s) | 127.27| 138.89 | 107.33 | 178.89 | 194.51 | 178.43 | 233.50 | 208.70 | 196.33 |
>
> Table II. Run time (s) for Chicago cases
> | Sim Duration (s) | 600 | 600 | 600 | 900 | 900 | 900 | 1200 | 1200 | 1200 |
> |--------|--------|--------|--------|--------|--------|--------|--------|--------|--------|
> | Cases | Case 1 | Case 2 | Case 3 | Case 4 | Case 5 | Case 6 | Case 7 | Case 8 | Case 9 |
> | Run Time (s) | 119.31 | 133.19 | 116.82 | 141.21 | 149.80 | 178.70 | 223.89 | 155.63 | 216.92 |
>
> **Comment 5**. I think the author(s) should clarify that all information contained in $S_t$ and $H_{t-1}$ positively contributes to the method's performance. Besides, could author(s) provide a specific example of $S_t$ and $H_{t-1}$?
>
> **Response**: Thanks for the reviewer's comments. We have clarified and expanded this explanation in the main body of the paper.  They are all basic inputs to LLM, and eliminating them will make it difficult for LLM to understand the current environmental traffic situation.
> - Current system state at time $t$: $S_t$ = {$V_t, P_t, C_t$}, where $V_t$ is the set of active vehicles, each with location, availability status and estimated arrival time. $P_t $ denotes the set of pending passenger requests, including pickup and drop-off locations and request times. $C_t $ is environment constraints, such travel time dynamics.
> - Historical Decision Trajectory up to $t-1$: $H_{t-1}$ = {$(a_0, f_0), (a_1, f_1), ..., (a_{t-1}, f_{t-1})$}, where $a_k$ is the LLM-generated high-level objective for time $k$ (not the dispatch/routing action obtained the optimization solver) based on $S_k$. $f_k$ is a scalar fitness score (average passenger delay) used to evaluate the effectiveness of $a_k$ via the downstream solver.
>
> Both components are critical in constructing the LLM prompt. $S_t$ provides essential real-time context, while $H_{t-1}$ enables the LLM to identify temporal patterns, such as repeated failures or regional imbalances. These form the dynamic prompt component ${P}_{dyn}$, which enables the LLM to adaptively refine the objective function based on evolving conditions.
> The following JSON illustrates the structure at step $t$ for dynamic prompt:
> `````
> {"taxi info": [ "Taxi 0: start_pos=24, est_arr_time=0s", ... "Taxi 59: start_pos=8, est_arr_time=100s"], "passenger info": [ "Passenger 1: origin=8, dest=32, requestT=181s", ... "Passenger 31: origin=5, dest=32, requestT=248s"],
>  "llm response": {"obj description": "New objective function focuses on minimizing passenger waiting time, taxi travel time, and balancing assignments with adjusted weights.", "obj code": "def dynamic_obj_func(self):\n  cost1 = ...\n  cost2 = ...\n  cost3 = ...\n  weights = [10, 20, 1000]\n  objective = sum(w*c for w,c in zip(weights, [cost1, cost2, cost3]))\n  self.model.setObjective(objective, gb.GRB.MINIMIZE)" }, "response format": "Your obj function is correct. Gurobi accepts your obj.", "evaluation score": 320}
> `````
> **Comment 6**. Is it possible to dynamically adjust the parameters $\alpha$, $\beta$, $\gamma$ to improve the performance of manual objectives? Moreover, the experiments do not specify the settings of these parameters.
>
> **Response**: Thanks for the reviewer's comments.  In our experiments, the weights of manual objective methods (as baseline methods at Tables 2 and 5 in the paper) strictly follow fixed values from prior work [2][4] (cited in the paper), ensuring consistency and fair benchmarking.
>
> The objective cost is defined as: $$obj = \sum_{i} weight_{i} \times cost_{i}$$
> Adjusting weights ($\alpha$, $\beta$, $\gamma$) is just one degree of freedom, also, modifying only the weights of the fixed cost does not yield much improvement. However, LLM can generate flexible weight and cost adjustments.
>
> Appendix 9.4 (line 959) in the paper provides several examples of high-level objectives generated by our LLM-guided method. As captured in Figures 17 and 18 in the paper, the model combines creative costs with novel dynamic and non-linear penalties that are challenging to define manually.
>
> **Comment 7**. Can multiple high-level objectives be pre-designed using LLMs and the appropriate objective be automatically selected at each time step? I think this could significantly reduce both runtime and the cost associated with using LLMs.
>
> **Response**: Thanks for the reviewer's suggestions. Pre-generating a library of high-level objectives can reduce LLM inference overhead, but in dynamic environments like real-time mobility systems, static objectives may quickly become suboptimal due to changing conditions (e.g., demand spikes or congestion). To maintain adaptability while ensuring low latency and cost efficiency, LLMs can be locally deployed and queried selectively for just-in-time updates. Furthermore, recent LLM advancements (e.g., quantized models and lightweight adapters) make it practical to update or fine-tune objective-generation logic periodically based on recent performance trends, achieving both responsiveness and efficiency.

---

> ### Comment · Reviewer_8CYJ · 2025-08-02
>
> My major concern is that every component of the method should adhere to the principle of Occam's razor. It is necessary to clarify why the current methods are inadequate or inapplicable, thereby motivating the proposal of a novel method. The simplest way to demonstrate this is through experimental evidence.
>
> ### About Comment 1
> Could the author(s) provide relevant evidence, such as existing studies or experiments?
>
> ### About Comment 2
> My intention is not to describe the specific tasks addressed by EoH and FunSearch in their original papers, but rather to emphasize that their frameworks possess broad applicability. For example, using EoH and FunSearch to generate mathematical objective functions.
>
> ### About Comment 3
> I want to see the effect of removing any single operator from Table 1, or using only one operator (instead of parameter tuning).
>
> ### About Comment 5
> Is it possible to provide experimental evidence demonstrating that all these inputs are necessary? This is similar to the point I raised in "About Comment 3."
>
> ### About Comment 6
> I want to see evidence supporting the claim that "modifying only the weights of the fixed cost does not yield much improvement." The evidence provided in Appendix 9.4 is indirect and seems to merely demonstrate the flexibility of the proposed method.

---

> > ### Author Response · Authors · 2025-08-03
> > **Response to Comment 1**
> >
> > **Q1. Comment 1. Could the author(s) provide relevant evidence, such as existing studies or experiments?**
> >
> > **Response**: We appreciate the reviewer’s concern. To address this, we have completed the experiments on one of the typical evolutional strategies, genetic algorithm (GA), to compare with our proposed method.
> >
> > The logic of GA is implemented in the gaUtils class, and its structure is illustrated below:
> > ```
> > Class gaUtils:
> > 	@classmethod
> > 	def mutate(arg):
> > 	# mutation operations
> > 	# call class “TermExtractor” to extract necessary segments
> > 	# then conduct mutations
> > 	…
> > 	@classmethod
> > 	def crossover(arg):
> > 	# crossover operation
> > 	# call class “TermExtractor” to extract necessary segments from two parent prompts
> > 	# then conduct crossover
> > 	…
> > ```
> > Below is a short description of the additional code that was implemented to make GA’s mutation and crossover operations compatible with our problem.
> > - **GA’s mutation**: The line where different cost terms are combined is first identified. The combinatorial weights are then perturbed to create a new mutated prompt.
> > - **GA’s crossover operations**: For both parent prompts, lines corresponding to each cost terms are first extracted. A pair of cost terms are then randomly selected and switched to create a new child prompt.
> >
> > The simulation results under 3 different problem scales are shown in Tables I and II, which compares average passenger delays (minutes) between our Harmony Search (HS)-based method and Genetic Algorithm (GA) in Manhattan and Chicago environments.
> >
> > Table I. Average passenger delay (min) under Manhattan case under GA and proposed method
> > |Methods| Small Case| Medium Case | Large Case |
> > |---------------------|--------|--------|--------|
> > | Genetic Algorithm        | 5.49   | 5.10  | 7.31  |
> > | Our proposed method | 1.55   | 2.59  |4.01  |
> >
> > Table II. Average passenger delay (min) under Chicago case under GA and proposed method
> > |Methods| Small Case| Medium Case | Large Case |
> > |---------------------|--------|--------|--------|
> > | Genetic Algorithm        | 9.15   | 17.76  |  24.55 |
> > | Our proposed method | 8.65   | 12.32  |15.37|
> >
> > Both Table I and Table II show that our proposed method consistently outperforms GA across both city cases and all problem scales. While GA are standard in evolutionary computation, they are suboptimal for LLM prompt evolution due to semantic disruptions from crossover/mutation, which often produce incoherent prompts misaligned with fitness scores, and the need for complex parsing logic to preserve grammar and context. In contrast, HS better suits this task by reusing complete past prompts (via HMCR) to maintain semantic integrity and applying minimal, localized edits (via PAR) for stable and context-aware prompt refinement.

---

> > ### Author Response · Authors · 2025-08-03
> > **Response to Comment 2**
> >
> > **Q2. Comment 2. My intention is not to describe the specific tasks addressed by EoH and FunSearch in their original papers, but rather to emphasize that their frameworks possess broad applicability. For example, using EoH and FunSearch to generate mathematical objective functions.**
> >
> > **Response**：We thank the reviewer for the clarification. We now understand that the emphasis is on the applicability of FunSearch and EoH to generate mathematical objective functions.
> >
> > While FunSearch and EoH offer broad frameworks for function discovery, they lack critical capabilities required for mathematical objective function generation in our solver-integrated system:
> > - **Lack of constraint-aware prompting**:
> > FunSearch and EoH do not include mechanisms for encoding solver-specific constraints (e.g., linearity, feasibility, MILP syntax) into the prompt generation process. As a result, they cannot guarantee that the generated functions comply with the strict requirements of mathematical optimization solvers.
> > ```
> > In contrast, our method integrates constraint templates directly into prompt design (see Appendix A.9), ensuring every generated prompt adheres to solver-compatible structure and constraints.
> > ```
> > - **Missing Feasibility-by-Design Mechanism**:
> > FunSearch and EoH lack built-in feasibility guarantees, and therefore they cannot embed domain-specific constraints (Eqs. (1g)–(1m)) during generation, in other words,
> > it is hard for them to avoid generating infeasible solutions.
> > ```
> > Our framework integrates an optimizer (see Figure 1 in the paper) that ensures feasibility by construction, maintaining solution validity throughout the evolutionary process.
> > ```
> > - **Absence of dynamic feedback mechanism**:
> > FunSearch and EoH operate in relatively static evaluation settings and lack mechanisms for continuous feedback from dynamic simulators. This prevents them from adapting prompt generation based on real-time system performance or simulation states.
> > ```
> > Our framework includes a Dynamic System Block (see Figure 1 in the paper) that continuously links the optimizer and simulator, enabling prompt refinement based on evolving environment feedback and feasibility results.
> > ```
> > Due to these limitations, it is difficult for FunSearch or EoH to directly generate valid and effective mathematical objective functions for use in constrained optimization problems. While adapting them to this context is possible in theory, it would require adding solver-aware constraint handling, feasibility checking, and dynamic feedback loops, which essentially builds a new framework beyond their original scope.

---

> > ### Author Response · Authors · 2025-08-03
> > **Response to Comment 3**
> >
> > **Q3. Comment 3.  I want to see the effect of removing any single operator from Table 1, or using only one operator instead of parameter tuning.**
> >
> >
> > **Response**: We thank the reviewer for the clarification. We have conducted additional ablation experiments to evaluate the effectiveness of each operator when activated individually. The results, summarized in Tables III and IV, report the average passenger delay under different problem scales for both Manhattan and Chicago cases.
> >
> > **WAY1 (Random Inference)** focuses on pure exploration by generating new objectives without leveraging any historical parent prompt. **WAY2 (Heuristic Improvement)** leverages parent prompts with instructions aimed at incremental refinements, such as temporal alignment and resource weighting. **WAY3 (Innovative Generation)** promotes paradigm-shifting innovations guided by more complex instructions that consider multi-horizon optimization and dynamic cost generation.
> >
> > Table III. Average passenger delay (min) in the Manhattan case under different single-operator activations and the proposed method
> > |Activated Operators| Small Case| Medium Case | Large Case |
> > |---------------------|--------|--------|--------|
> > | WAY1        | 2.86   | 2.79  | 11.81 |
> > | WAY2        | 3.58   | 3.93  |7.28  |
> > | WAY3        | 1.60   | 4.57  |5.92 |
> > | Our proposed method | 1.55   | 2.59  |4.01  |
> >
> > Table IV. Average passenger delay (min) in the Chicago case under different single-operator activations and the proposed method
> > |Activated Operators| Small Case| Medium Case | Large Case |
> > |---------------------|--------|--------|--------|
> > | WAY1        | 20.89   | 13.94  | 18.92  |
> > | WAY2        | 9.74   | 19.06  |20.01  |
> > | WAY3        | 12.85   | 25.91  |20.80  |
> > | Our proposed method | 8.65   | 12.32  |15.37|
> > ```
> > These results highlight that while individual operators offer varying degrees of performance improvement, the combination implemented in our proposed method consistently yields the lowest average passenger delay across all cases.
> > ```
> > Way1 performs relatively well on smaller scales but struggles significantly on larger problems (especially in Manhattan large case and unbalanced distributed Chicago scenarios), indicating limited robustness. Way2 tends to produce moderate performance, often better than WAY1 in some medium cases but still lagging behind the full method, particularly in the large-scale cases. Way3 yields the best performance among the single-operator activations in Manhattan small case and shows some improvement in larger cases. However, it is inconsistent in the Chicago case, where delays remain high.
> >
> > The result indicates that relying on any single operator limits the performance and robustness of the HS algorithm. The combination of three operators systematically balances exploration and exploitation, leading to significant improvements in passenger wait times.

---

> ### Author Response · Authors · 2025-08-03
> **Response to Comment 5 and 6**
>
> **Q4. Comment 5. Is it possible to provide experimental evidence demonstrating that all these inputs are necessary? This is similar to the point I raised in "About Comment 3."**
>
> **Response**: We thank the reviewer for the clarification. To evaluate the impact of incorporating dynamic information in the prompt, we conducted ablation experiments by removing the state information ($S_{t}$) and historical objectives with fitness scores ($H_{t}$), collectively referred to as $P_{dyn}$. The modified prompt was then used to rerun simulations under the same settings. The results are summarized in Tables V and VI.
>
> Table V. Average passenger delay (min) under Manhattan case
> |Activated Operators| Small Case| Medium Case | Large Case |
> |---------------------|--------|--------|--------|
> | Without $P_{dyn}$ info      | 2.96    | 5.11  |19.47 |
> | Our proposed method | 1.55   | 2.59  |4.01  |
>
> Table VI. Average passenger delay (min) under Chicago case
> |Activated Operators| Small Case| Medium Case | Large Case |
> |---------------------|--------|--------|--------|
> | Without $P_{dyn}$ info       | 11.04   | 20.89  | 25.36 |
> | Our proposed method | 8.65   | 12.32  |15.37|
> ```
> These results demonstrate that excluding dynamic information (Pdyn) leads to significantly higher passenger delays, especially as the problem scale increases. This highlights the critical role of both state and historical context in guiding the generation of high-quality solutions in our framework.
> ```
> **Current system state $S_t$** captures essential real-time traffic and resource conditions, including the locations, availability, and estimated arrival times of active vehicles, pending passenger requests with origin/destination and request times, and dynamic environmental constraints such as travel time variations. Without $S_t$, the LLM lacks direct knowledge of the current operational context, making it difficult to design objectives to actual system status.
>
> **Historical decision trajectory $H_{t-1}$** contains prior LLM-generated high-level objectives paired with scalar fitness scores (average passenger delay), providing feedback on past performance. This historical record allows the LLM to identify temporal patterns, recurring failures, and regional imbalances, enabling refinement of objectives through learning from previous outcomes.
>
> **Q5. Comment 6. I want to see evidence supporting the claim that "modifying only the weights of the fixed cost does not yield much improvement." The evidence provided in Appendix 9.4 is indirect and seems to merely demonstrate the flexibility of the proposed method.**
>
> **Response**: Thanks for the reviewer's thoughtful and constructive feedback. We have conducted additional experiments to assess the impact of manually assigning different weight configurations to the objective cost components.
>
> We evaluated a range of weight combinations, including **weight-dominated settings**  (e.g., (100, 1, 1)) and **extremely imbalanced distributions** (e.g., (0,0,1) as third cost component shows good performance when weights are high) in both Manhattan and Chicago environments.
>
> Table VII. Average passenger delay (min) under Manhattan case with different weight combinations
> |( α,β,γ)| Small Case| Medium Case | Large Case |
> |---------------------|--------|--------|--------|
> | (100, 1, 1)         | 6.60   | 27.90  | 49.74  |
> | (1, 100, 1)         | 7.00   | 12.34  |44.06  |
> | (1, 1, 100)         | 3.32   | 6.06   | 9.27   |
> | (0, 0, 1)        | 13.10  | 13.64  | 16.38  |
> | Our proposed method | 1.55   | 2.59  | 4.01   |
>
> Table VIII. Average passenger delay (min) under Chicago case with different weight combinations
> | (α,   β,   γ)           | Small Case | Medium Case | Large Case |
> |---------------------|--------|--------|--------|
> | (100, 1, 1)         | 15.85  | 76.50  | 108.91 |
> | (1, 100, 1)         | 24.13  | 88.31  | 85.17  |
> | (1, 1, 100)         | 9.54   | 16.71  | 25.01  |
> | (0, 0, 1)        | 9.64   | 13.69  | 18.34  |
> | Our proposed method | 8.65   | 12.32  | 15.37  |
> ```
> We can find from the above two tables that: Our approach consistently achieves the lowest delays across all scenarios in both environments. Also, high γ weights (1,1,100) significantly reduce delays compared to α or β dominated weights. However, the (0,0,1) edge case underperforms (1,1,100), suggesting that moderate contributions from α and β components are still necessary for optimal outcomes.
> ```
> Note that our prior baseline already emphasized the third component (**(1,1,100) is adopted in baseline experiment at Tables 2 and 5 in the paper**) to promote better taxi utilization. In contrast, our proposed method leverages dynamic, context-aware prompt generation, which adaptively adjust both weights and costs.

---

> > ### Comment · Reviewer_8CYJ · 2025-08-04
> >
> > Thank you for your response. Most of my concerns have now been satisfactorily addressed, so I have increased my rating. I appreciate the effort involved in preparing additional results during the rebuttal period. However, I strongly recommend that the author(s) include a more comprehensive and robust empirical study in the camera-ready version (if accepted), with additional experiments to demonstrate how the new method improves performance. I believe these results will interest a broad audience.

---

### Official Review · Reviewer_ap7C · 2025-07-03

**Clarity:** 3
**Significance:** 3
**Originality:** 4
**Rating:** 5
**Confidence:** 2

**Summary:**

This paper proposes an LLM-integrated hierarchical optimization framework for mobility-on-demand systems. Specifically, an LLM is employed to generate the objective function for the assignment problem, where the harmony search algorithm iteratively refines the generation prompts. The optimization problems at the first and second levels are then solved numerically. The proposed framework achieves higher performance compared to baseline approaches, including methods based on manually designed objectives, RL-based methods, and purely LLM-based methods, when evaluated on real-world taxi datasets.

**Questions:**

- Could smaller language models be fine-tuned to generate effective objective functions, even with fixed prompts? What is the main advantage of the proposed evolutionary framework compared to fine-tuning?
- Would it be impossible to design effective objective functions using rule-based methods? Is the use of LLMs necessary for generating high-quality objectives? How complex are the behaviors demonstrated by LLMs in the experiments?

**Ethical Concerns:**

["NO or VERY MINOR ethics concerns only"]

**Final Justification:**

While the motivation behind the harmony search and each component of the framework is not very clear, and the scalability of the algorithm has not been clearly demonstrated, I think the proposed framework is promising and a meaningful contribution. I will keep my score.

**Limitations:**

Yes.

**Paper Formatting Concerns:**

No formatting concerns found.

**Quality:**

4

**Strengths And Weaknesses:**

**Strengths:**
- This work proposes a novel method for integrating LLMs with optimization problems that can leverage any pre-trained LLM. There is considerable room to further enhance the proposed framework by fine-tuning the LLM.
- The proposed framework is described clearly and thoroughly, making it easy to reproduce.
- The empirical evaluations are comprehensive and promising, demonstrating that the proposed framework generally outperforms the baseline methods.

**Weaknesses:**
- The proposed algorithm relies on multiple heuristics, making it challenging to clearly evaluate how each component individually contributes to the overall performance.

---

> ### Author Rebuttal · Authors · 2025-07-30
>
> **We appreciate the reviewers’ thorough feedback. We have grouped both the identified weaknesses and questions into the following comments for clarity and response.**
>
> **Comment 1**. The proposed algorithm relies on multiple heuristics, making it challenging to clearly evaluate how each component individually contributes to the overall performance.
>
> **Response**: Thanks for the reviewer's comments. We appreciate the reviewer’s insightful observation about the challenge of individual component contributions.
>
> The integrated design is essential to address the inherent complexity of dynamic ride-hailing dispatch. Specifically, the environment is highly stochastic, partially observable, and multi-objective in nature, which makes end-to-end training or rule-based optimization alone insufficient. A hierarchical approach is thus required, where separate components handle specific challenges but operate together seamlessly:
> - The **high-level module**, aided by the *LLM*, is responsible for generating interpretable and flexible dispatch objectives that adapt to evolving traffic and passenger patterns.
> - The **low-level routing solver** ensures that these objectives are translated into executable dispatch plans that satisfy hard constraints.
> - The **harmony search loop** acts as a *meta-heuristic* that iteratively refines the prompts to query LLM using simulation feedback, driving the co-adaptation between the semantic layer and the optimization layer.
>
> Our system modules fit together seamlessly, and modules are interdependent. For example, the LLM alone cannot account for downstream feasibility without the routing feedback, and the routing solver depends on the quality of high-level objectives to make meaningful decisions. In this sense, the LLM and harmony search together form a co-evolving meta-heuristic, and isolating them would undermine the core mechanism of design.
>
> We believe our empirical results offer strong indirect evidence for the effectiveness of each component. Extensive experiments across diverse scenarios based on both New York and Chicago taxi datasets, demonstrate **an average reduction of 16% on passenger waiting times** over state-of-the-art baselines (see Table 2 and Table 5 in the paper). Specifically,
> - In Manhattan testing, our approach consistently achieves ~40\% lower waiting times across large-scale scenarios, e.g., for Case 9 in Table 2 in the paper, our approach is 4.01 min compared to RL’s 6.82 min.
> - Under large demand case, FunSearch completes all pickups in 5–7 slots in Manhattan and 6 slots in Chicago, while RL and manual baselines need 8/9 slots in the latter (see Figures 7 and 8 in the paper Appendix). However, our method consistently fulfills all requests in just 4 slots (2400s in Manhattan, 3200s in Chicago), achieving 40–50\% faster completion.
>
> **Comment 2**. Could smaller language models be fine-tuned to generate effective objective functions, even with fixed prompts? What is the main advantage of the proposed evolutionary framework compared to fine-tuning?
>
> **Response**: Thanks for the reviewer's comments. While fine-tuning smaller LLMs for objective generation is possible, our evolutionary framework offers critical advantages for dynamic ride-hailing systems where traffic conditions, passenger requests, and driver positions change continuously. Below we clarify why fixed-prompt approaches are insufficient and explain the need to use the proposed evolutionary framework:
> - **Dynamic adaptability to system state changes**: Fixed prompts or fine-tuned models without feedback loop generate static outputs that cannot evolve in response to real-time changes in the ride-hailing system. Our framework uses feedback-driven prompt evolution via Harmony Search to refine the LLM-generated objective functions iteratively, ensuring alignment with the current state of the ride-hailing system.
> - **Data/compute efficiency**: Fine-tuning requires extensive data, computational resources, and retraining when domain shifts occur, e.g., new cities, different demand patterns. Our method is much quicker and more efficient as we only need to evolve the prompt, guided by the downstream optimization feedback.
> - **Mitigation of partial observability**: Fine-tuned model with fixed prompts lack full foresight of the environment dynamics, leading to misalignment. By using LLM as a meta-objective designer within a closed-loop evolutionary process, we overcome the partial observability challenge and reduce reliance on human-crafted heuristics.
>
> Therefore, while fine-tuning small LLMs can yield good performance in static or narrowly scoped settings, it is insufficient to address the dynamic and real-time decision-making challenges inherent in ride-hailing dispatch systems. In contrast, our proposed framework, embedding LLMs within an evolutionary optimization loop, enables **real-time adaptivity**, **training-free generalization**, and **solver-grounded solution quality**, which are essential for practical deployment in large-scale urban mobility systems.
>
> **Comment 3**. Would it be impossible to design effective objective functions using rule-based methods? Is the use of LLMs necessary for generating high-quality objectives? How complex are the behaviors demonstrated by LLMs in the experiments?
>
> **Response**: Thanks for the reviewer's comments. While rule-based objectives are feasible, our experiments reveal that traditional designs often rely on fixed linear/quadratic penalties or manually tuned weights. Therefore, it is difficult to adapt dynamically to complex systems like ride-hailing, where passenger requests, traffic, and taxi availability vary rapidly over time. Also, rule-based methods often rely on extensive domain expertise and can only be modified under manual recalibration.
>
> An example of a generated objective from our experimental results is provided below:
> `````
> def dynamic_obj_func(self):
>       # Improved objective function to enhance taxi utilization by considering arrival times, dynamic weights, and better load  balancing.
>         print("Creating dynamic objectives for Assignment Model")
>
>         # Cost component 1: Passenger waiting time from request to taxi arrival
>         cost1 = gb.quicksum(
>             self.y[v,p] * max(self.taxi[v].arrival_time - self.passenger[p].arrTime, 0)
>             for v in self.taxi.keys() for p in self.passenger.keys())
>
>         # Cost component 2: Taxi detour time from current position to passenger pickup
>         cost2 = gb.quicksum(
>             self.y[v,p] * self.distMatrix[self.taxi[v].start_pos][self.passenger[p].origin]
>             for v in self.taxi.keys() for p in self.passenger.keys())
>
>         # Cost component 3: Passenger trip time from pickup to drop-off
>         cost3 = gb.quicksum(
>             self.y[v,p] * self.distMatrix[self.passenger[p].origin][self.passenger[p].destination]
>             for v in self.taxi.keys() for p in self.passenger.keys())
>
>         # Cost component 4: Penalty for multiple assignments per taxi
>         cost4 = gb.quicksum(
>             (gb.quicksum(self.y[v,p] for p in self.passenger.keys())) ** 2
>             for v in self.taxi.keys())
>
>         # Calculate dynamic weights based on idle taxis
>         idle_taxis = sum(
>             1 for v in self.taxi.values()
>             if v.arrival_time <= self.passenger[next(iter(self.passenger.keys()))].arrTime)
>
>         weight_factor = 1 + (idle_taxis / len(self.taxi) if len(self.taxi) else 0)
>
>         # Cost component 5: Load balancing penalty across taxis
>         load_balance_factor = 1.5 * weight_factor
>         cost5 = gb.quicksum(
>             (gb.quicksum(self.y[v,p] for p in self.passenger.keys()) -
>              (len(self.passenger) / len(self.taxi))) ** 2 * load_balance_factor
>             for v in self.taxi.keys())
>
>         # Adjust weights for better taxi utilization
>         weights = [
>             weight_factor * 2.5,   # Higher weight on passenger waiting time
>             weight_factor * 1.75,  # Consider taxi detour time
>             weight_factor * 1.25,  # Ensure efficient passenger trips
>             1200 * weight_factor,  # Strong penalty for overloading taxis
>             300 * weight_factor    # Enhanced load balancing importance
>         ]
>
>         # Combine cost components
>         costs = [cost1, cost2, cost3, cost4, cost5]
>
>         # Set objective
>         objective = sum(w * c for w, c in zip(weights, costs))
>         self.model.setObjective(objective, gb.GRB.MINIMIZE)
> `````
>
> Clearly, the LLM-generated objective (as shown above) has multiple cost components with dynamic weights that adjust based on system states, which is beyond typical rule-based design. It effectively captures trade-offs among passenger wait time, detour time, and load balancing, adapting in real time to system conditions. Our experiments (see Tables 2 and 5 in the paper) validate its effectiveness: In the Manhattan downtown scenario, especially large cases, our method achieves delay costs that are approximately **∼40% lower** than the best-performing rule-based method, aligning with the final drop-off time trends in Figures 7 and 8 in the paper Appendix.

---

> > ### Comment · Reviewer_ap7C · 2025-08-04
> >
> > Thank you for the detailed response and I will keep my original score.

---

### Official Review · Reviewer_7QZG · 2025-07-14

**Clarity:** 3
**Significance:** 3
**Originality:** 3
**Rating:** 3
**Confidence:** 3

**Summary:**

This paper proposes a training-free hierarchical optimization framework for mobility-on-demand systems, where a large language model (LLM) dynamically generates high-level assignment objectives. These are refined via a harmony search-based evolutionary loop and solved with a low-level optimizer for routing under constraints. Experiments on real-world taxi datasets show significant improvements over RL and LLM-only baselines.

**Questions:**

1. Can authors provide more clear movitation for selecting HS, instead of other ES? (see Cons 1)
2. Can authors provide experiment results over large scale of taxis and passengers, for example, thousands of taxis? (see Cons 2)

**Ethical Concerns:**

["NO or VERY MINOR ethics concerns only"]

**Final Justification:**

I have thoroughly read the rebuttals from the reviewers, and part of my concerns have been resolved, and I will keep my score

**Limitations:**

"yes"

**Quality:**

3

**Strengths And Weaknesses:**

Pros:

1. The integration of LLMs and optimization in this paper is novel.
2. The training-free scheme is practical, which requires no training data comprared with RL-based approaches.
3. The proposed approach is evaluated on large-scale, real-world taxi datasets, demonstrating consistent and significant improvements over SOTAs.



Cons:

My major concerns include **the introduction of HS**, and **the scalability of method**.

1. The motivation for using HS is weak, and the paper does not explore alternative evolutionary strategies to justify this choice.
2. The problem scale seems unrealistic, as real-world ride-hailing systems typically involve thousands of taxis, while the experiments are limited to hundreds.
3. The framework relies on frequent LLM queries, but the runtime cost and inference latency in real-time deployment are not analyzed.
4. Figure 1 is overly complex and unclear. the difference between WAY 2 and WAY 3 is not visually or conceptually well-distinguished.

---

> ### Author Rebuttal · Authors · 2025-07-30
>
> **We appreciate the reviewers’ thorough feedback. We have grouped both the identified weaknesses and questions into the following comments for clarity and response.**
>
> **Comment 1**. The motivation for using HS is weak, and the paper does not explore alternative evolutionary strategies to justify this choice. Can authors provide more clear motivation for selecting HS, instead of other ES?
>
> **Response**: Thanks for the reviewer's comments. Our selection of HS is grounded in the unique challenges of evolving LLM prompts, which encode rich problem context, including environment state $S_t$ and historical llm-generated objectives $H_t$ in natural language. Meanwhile, traditional evolutionary strategies, such as Genetic Algorithms (GA), typically rely on crossover and mutation operations at the individual solution level. These operations suffer from several fundamental limitations when working with LLM prompts:
>
> 1. ***Loss of Coherence***: GA’s crossover operation between arbitrary prompt segments often results in semantically inconsistent text, making it difficult for the LLM to understand or generate meaningful responses.
> ```
> In contrast, the HS algorithm operates on complete prompts using operators that preserve natural language structure. It either generates new prompts from scratch or reuses full parent prompts, avoiding disruptive operations like crossover and mutation.
> ```
> 2. ***Fitness Inconsistency***: In our case, GA’s crossover and mutation operations alter the structure of a prompt, breaking the link between its phrasing and the LLM's response quality. More specifically, modifying LLM-generated objective functions (e.g., weights or costs) disrupts the alignment between prompts and their fitness scores, making parent fitness values unreliable for guiding future search.
> ```
> HS avoids this issue by either generating new prompts without relying on prior fitness scores to promote diversity, or directly reusing full parent prompts with their known fitness values.
> ```
> 3. ***Parsing Complexity***: To perform crossover and mutation correctly, additional parsing logic is required to identify and recombine meaningful prompt segments, increasing implementation complexity and reducing algorithmic robustness.
> ```
> Since HS does not use crossover or mutation, it eliminates the need for prompt parsing and maintains a simpler, more robust design.
> ```
> To summarize, HS is selected due to its advantages on maintaining the structural and semantic constraints, which is required for stable LLM interaction.
>
> **Comment 2**. The problem scale seems unrealistic, as real-world ride-hailing systems typically involve thousands of taxis, while the experiments are limited to hundreds. Can authors provide experiment results over large scale of taxis and passengers, for example, thousands of taxis?
>
> **Response**: Thanks for the reviewer's comments. We conducted additional large-scale experiments that extend our previous settings. Specifically, we scaled our simulations to include up to 1000 taxis and 1000 passengers within the Manhattan and Chicago environments, under longer simulation durations.
>
>     The results are summarized in Table I. Consistent with the small-scale findings in Tables 2 and 5 in the paper, the Manhattan scenarios in large cases below consistently exhibit lower average passenger wait times compared to Chicago, owing to their more spatially balanced demand distribution. Conversely, the more skewed distribution in the Chicago scenarios leads to longer delays under high demand levels. These results demonstrate that our method maintains solution quality and computational tractability even at significantly larger problem scales.
>
> Table I. Experiment for scenarios under 1hour simulation period
> | Scenarios | Manhattan | Manhattan | Manhattan | Manhattan | Chicago | Chicago | Chicago | Chicago |
> |--------|--------|--------|--------|--------|--------|--------|--------|--------|
> | Taxi No. | 300 | 800 | 1000 | 500 | 300 | 800 | 1000 | 500 |
> | Passenger No. | 500 | 500 | 800 | 1000 | 500 | 500 | 800 | 1000 |
> | Avg. Wait time (min) | 3.69 | 0.47 | 1.82 | 4.64 | 5.21 | 1.02 | 13.65 | 6.23 |
>
> Moreover, for real-world deployments involving tens of thousands of agents, our framework can be extended through a distributed architecture. In such settings, multiple instances of our controller can operate in parallel across different city zones, with coordination mechanisms ensuring global consistency and service quality.
>
> **Comment 3**. The framework relies on frequent LLM queries, but the runtime cost and inference latency in real-time deployment are not analyzed.
>
> **Response**: Thanks for the reviewer's comments. We have provided the actual runtime statistics for the optimization & prompt evolution components across the Manhattan and Chicago cases.
> ```
> As shown in Table II and Table III, although runtime increases with the simulation scale (e.g., longer durations and more demand), it remains substantially lower than the simulation time itself. For instance, Case 9 in Table II, which involves a 1200s (20min) simulation, completes in less than 4min, even though it includes multiple rounds of optimization. This demonstrates that the framework can operate within practical time constraints.
> ```
> Table II. Run time (s) for Manhattan cases
> | Sim Duration (s) | 600 | 600 | 600 | 900 | 900 | 900 | 1200 | 1200 | 1200 |
> |--------|--------|--------|--------|--------|--------|--------|--------|--------|--------|
> | Cases | Case 1 | Case 2 | Case 3 | Case 4 | Case 5 | Case 6 | Case 7 | Case 8 | Case 9 |
> | Run Time (s) | 127.27| 138.89 | 107.33 | 178.89 | 194.51 | 178.43 | 233.50 | 208.70 | 196.33 |
> ```
> Similar to Manhattan case,  Chicago case (Table III below) run times grow moderately with the problem scale, yet all scenarios are solved within 4 minutes, demonstrating practical feasibility.
> ```
> Table III. Run time (s) for Chicago cases
> | Sim Duration (s) | 600 | 600 | 600 | 900 | 900 | 900 | 1200 | 1200 | 1200 |
> |--------|--------|--------|--------|--------|--------|--------|--------|--------|--------|
> | Cases | Case 1 | Case 2 | Case 3 | Case 4 | Case 5 | Case 6 | Case 7 | Case 8 | Case 9 |
> | Run Time (s) | 119.31 | 133.19 | 116.82 | 141.21 | 149.80 | 178.70 | 223.89 | 155.63 | 216.92 |
>
> **Comment 4**. Figure 1 is overly complex and unclear. the difference between WAY 2 and WAY 3 is not visually or conceptually well-distinguished.
>
> **Response**: Thanks for the reviewer's comments. We appreciate the reviewer’s observation regarding the lack of clarity in differentiating WAY 2 and WAY 3 in Figure 1. The original design aimed to present a unified framework encompassing three evolutionary strategies:
> - WAY 1: pure exploration from scratch.
> - WAY 2: exploitation via incremental refinement of elite prompts.
> - WAY 3: exploitation via structural recombination and reinvention.
>
> While both WAY 2 and WAY 3 share downstream optimization components to emphasize architectural cohesion, the visual simplification unintentionally obscured their conceptual distinctions. To address this, we have revised Figure 1 with the following modifications:
> - *Visual Differentiation*: Label WAY 2 as “Incremental Refinement”, using a red block with dashed borders to denote constraint-aware local adjustments. Label WAY 3 as “Paradigm Innovation”, using a green block with double borders to signal structural novelty via recombination.
> - *Instruction Mapping*: The figure should clearly include the main prompt-level mechanisms. For WAY 2, these are temporal alignment, resource weighting, and preserving partial existing parent semantics. For WAY 3, the focus is on long-horizon impacts, increasing the use of idle taxis and innovative goals.
> - *Structural Simplification*: Collapse shared modules into a single “Shared Optimization Layer” to reduce visual redundancy. Differentiate data flow: solid arrows for WAY 2, zig-zag for WAY 3.

---

> > ### Comment · Reviewer_7QZG · 2025-08-07
> >
> > Thanks for your comment, most of my concerns have been resolved

---

> > > ### Author Response · Authors · 2025-08-07
> > >
> > > Dear Reviewer,
> > >
> > > Thank you for your follow-up and for taking the time to review our work. We're glad to hear that most of your concerns have been resolved. If there are any remaining issues or suggestions for improvement that we may have missed, we would greatly appreciate it if you could kindly let us know. We're happy to further clarify or address any outstanding points.
> > >
> > > Best Regards,
> > > The Authors

---

### Official Review · Reviewer_E53E · 2025-07-16

**Clarity:** 3
**Significance:** 3
**Originality:** 2
**Rating:** 4
**Confidence:** 4

**Summary:**

This paper introduces a hybrid framework that combines large language models (LLMs) with mathematical optimization to address dynamic decision-making, specifically for mobility-on-demand systems. The approach uses pre-trained LLMs to generate high-level objectives and employs mathematical optimization for detailed dispatching strategies, creating a system that doesn't require extensive training.

The method integrates LLM-driven semantic reasoning with solver-enforced mathematical rigor, dynamically updating objectives through prompts. This closed-loop mechanism balances exploration and feasibility, achieving state-of-the-art performance on urban mobility benchmarks. Experiments on New York and Chicago taxi datasets demonstrate the system's effectiveness in minimizing delays and maintaining spatial consistency. Detailed analyses, including ablation studies and hyperparameter sensitivity, provide comprehensive evaluation of the framework's components.

Overall, the paper presents a creative integration of NLP and optimization methods, offering a new approach to complex real-time decision-making. By showcasing adaptability across different urban environments and demand patterns, the authors highlight practical applications and future research potential in combining data-driven priors with mathematical optimization for robust decision-making systems. The thorough experimental setup ensures reproducibility and extension of this promising approach.

**Questions:**

1、How do you plan to incorporate more dynamic factors, such as traffic congestion or road disruptions into your model? Could you elaborate on potential strategies or experiments that might address these limitations?

2、Can you provide a deeper comparison with existing state-of-the-art techniques, highlighting both the advantages and limitations of your approach relative to others?

3、Could you explain further how the LLM adapts through prompt-level feedback? Are there any plans to integrate reinforcement learning techniques to improve adaptability?

**Ethical Concerns:**

["NO or VERY MINOR ethics concerns only"]

**Final Justification:**

Thank you for your detailed response, and most of my concerns have been solved.

**Limitations:**

The authors have partially addressed the limitations of their work, particularly regarding technical constraints such as the static adaptation mechanism of the LLM and the lack of fine-grained traffic dynamics in the current experimental setup. However, they have not sufficiently discussed potential negative societal impacts or broader ethical considerations associated with deploying such a system in real-world, high-stakes environments.

For instance, the use of LLM-generated prompts in decision-making systems could introduce biases or unintended consequences, especially if the generated objectives inadvertently prioritize certain user groups or regions over others. Additionally, overreliance on automated dispatching systems without proper human oversight may lead to safety concerns or reduced job control for drivers in mobility-on-demand platforms.

**Paper Formatting Concerns:**

There are no major format issues

**Quality:**

2

**Strengths And Weaknesses:**

Strengths：
The paper stands out for its high quality and clarity, offering a well-documented experimental setup that ensures reproducibility. The integration of large language models with mathematical optimization techniques is clearly explained, making the methodology accessible to both experts and non-experts. By demonstrating practical applications on real-world datasets from New York and Chicago taxi services, the authors highlight the significant potential of their approach in urban mobility systems. This combination not only enhances the accuracy of predictions but also provides confidence intervals crucial for decision support systems.

Weaknesses：
Despite its strengths, the paper has some limitations, particularly regarding the applicability of its model in more realistic operational conditions. The current setup does not fully account for dynamic factors such as traffic congestion or road disruptions, which could limit the model's effectiveness in real-world scenarios. Additionally, while the proposed method is innovative, a deeper comparison with existing state-of-the-art techniques is lacking, leaving gaps in understanding its unique advantages and potential areas for improvement. Addressing these aspects would strengthen the paper’s overall impact and broaden its applicability.

---

> ### Author Rebuttal · Authors · 2025-07-30
>
> **We appreciate the reviewers’ thorough feedback. We have grouped both the identified weaknesses, questions and limitations into the following comments for clarity and response.**
>
> **Comment 1**. How do you plan to incorporate more dynamic factors, such as traffic congestion or road disruptions into your model? Could you elaborate on potential strategies or experiments that might address these limitations?
>
> **Response**: Thanks for the reviewer's comments. Our work establishes a hybrid LLM $+$ optimizer framework that combines high-level reasoning with numerical optimization for dynamic decision-making. The LLM serves as a meta-optimizer to evolve high-level objectives, while a mathematical solver ensures feasibility and rigor. Experiments conducted under New York and Chicago datasets achieve an average reduction of 16% on passenger waiting time, demonstrating that our approach can adapt to different environment dynamics.
>
> To incorporate more dynamic factors and make the test scenarios even more realistic, we plan to extend our framework along the directions below:
>
> **Enhanced Traffic Simulation**: Our current setup already uses real-world traffic data to derive empirical distributions for realistic taxi locations and passenger request times, capturing the stochastic nature of urban mobility. Future work will further enrich the simulation by incorporating broader traffic patterns, including non-taxi vehicles and junction signals.
>
> **What-if Scenario Analysis**: We will conduct experiments to assess how the proposed method responds to unexpected events, such as accidents, road blockages, and others. This stress testing will help evaluate robustness of the LLM-guided optimization process.
>
> **Utilization of Advanced Traffic Simulator**: We will replace our current simulator class with established traffic simulation platforms like SUMO or VISSIM, which support dynamic vehicle interactions, signal control, and detailed road network modeling.
>
> We believe that these improvements, aimed at incorporating more dynamic elements, will allow us to simulate complex urban scenarios with realistic congestion patterns and road disruptions. Therefore, we can gain insight into how AI-generated targets perform in dynamic, uncertain transportation environments.
>
> **Comment 2**. Can you provide a deeper comparison with existing state-of-the-art techniques, highlighting both the advantages and limitations of your approach relative to others?
>
> **Response**: Thanks for the reviewer's comments. We provide the advantages and limitations of our hybrid LLM$+$optimizer approach against state-of-the-art baselines as follows, supported by empirical results from Tables 2 and 5, and spatiotemporal analysis (Appendix A.1.1) in the paper.
>
> **Methodological Advantages of Our Hybrid Approach**:
> - To the best of our knowledge, this is the **first work to combine LLM-driven semantic reasoning with classical mathematical optimization** in a closed-loop framework for sequential, dynamic decision-making.
> - Unlike existing baselines, such as manually-designed objective functions, RL-based methods, or pure LLM-guided solvers (e.g., FunSearch, EoH), our framework is **training free**, and allows the **LLM to generate high-level objectives** via prompt-based reasoning, while a **solver guarantees constraint satisfaction and numerical rigor**.
> - We propose a **novel harmony search algorithm** with three operators (random inference, heuristic improvement, innovative generation) that iteratively refine LLM-generated objectives based on solver feedback.
>
> This mechanism of LLM-guided objective refinement through solver feedback is not present in any state-of-the-art approaches.
>
> **Empirical Advantages of Our Hybrid Approach**:
> - *Superior Scalability and Solution Consistency*: Manual designed objectives (e.g., weighted distance-temporal-utilization terms) suffer from significant performance degradation at scale, e.g., 9.27 min for case 9 in Manhattan scenario in Table 2. However, our approach consistently achieves ~40\% lower waiting times across large-scale scenarios, e.g., Ours: 4.01 min vs. RL: 6.82 min (best state-of-the-art) for case 9 in Table 2.
>
> - *Robustness to Spatiotemporal Heterogeneity*: Manual objectives, RL and LLM-only methods fail in imbalanced distributions, e.g., Chicago case 9, RL yields 20.03 min delay in Table 5. However, our approach reduces waiting times by ~20\% in Chicago’s large-scale, unbalanced zones, e.g., case 9: 15.37 min vs. EoH: 19.79 min.
>
>  **Current Limitations**:
>
> - *Low-Demand Period Overhead*: During low-demand periods (e.g., nighttime), our system still operates with full objective generation and solver cycles, which may be unnecessary. This can be mitigated by time-segmented deployment, using simpler objectives during off-peak hours, while applying our full method during peak periods.
>
> - *LLM Infrastructure Requirements*: Compared to rule-based methods, our method requires LLM infrastructure and proper prompt engineering. However, this cost becomes negligible in large-scale deployments and can be further reduced by using locally hosted models or lightweight techniques.
>
> **Comment 3**. Could you explain further how the LLM adapts through prompt-level feedback? Are there any plans to integrate reinforcement learning techniques to improve adaptability?
>
> **Response**: Thanks for the reviewer's comments.
>
> **How the LLM adapts through prompt-level feedback**
>
> Our approach integrates Harmony Search (HS), multi-query inference, and dynamic state streaming to facilitate real-time LLM adaptation within a closed-loop control framework.
>
> At each simulation timestep $t$, we execute closed-loop prompting, where the LLM receives structured feedback reflecting the current system state and generates updated dispatch objectives accordingly. The dynamic prompt $\mathcal{P}_{\text{dyn}}$ includes two core components:
>
> - *Vehicle states*: $\text{veh}_t \in \mathbb{R}^{|\text{veh}| \times 2}$, real-time positions and expected arrival times.
> - *Passenger demand tensor*: $\text{pass}_t \in \mathbb{R}^{|\text{pass}| \times 3}$, origin, destination, and request timestamps.
>
> HS is used to optimize prompt structures across generations. Multi-query inference enables iterative refinement based on optimizer feedback. Dynamic prompting ensures the LLM remains contextually grounded in the evolving simulation environment. Together, these components form a co-adaptive feedback loop between the LLM and the optimization process.
>
> This integrated mechanism is validated in Table 4 and Figure 9 in the paper. In particular, we compare multi-query (closed-loop) against single-query (open-loop) strategies. The results show that multi-query consistently achieves up to **50\% lower cost** within 10 iterations. Notably, the steepest cost reductions occur in the early iterations, demonstrating the efficacy of prompt-level adaptation in accelerating convergence.
>
> **RL-based plan**
>
> We agree this is a worthwhile direction and is a potential extension to our framework. Unlike the baseline RL methods evaluated in our paper, this extension will emphasize the co-adaptation between the LLM and the optimizer for enhanced performance.
>
> While baseline RL methods directly learn low-level dispatch or routing policies, our planned RL integration treats the RL algorithm as a tuner that improves the LLM’s ability to generate high-level, interpretable objective functions using feedback from the optimizer as a critic. Specifically,
> - *LLM* acts as *actor* to generate objective functions.
> - *Optimizer* acts as *critic* to evaluate objectives by solving a mixed-integer programming problem. The output $W$ of the optimizer is used to update RL rewards.
> - *Reward*: $\mathcal{R} = -\Delta W + \lambda \cdot \text{Explainability Score}$, where $\Delta W$ is the negative change in cost (e.g., passenger waiting time, $W_{k+1} - W_{k}$) and $\lambda \cdot \text{ExplainabilityScore}$ is the weighted human rating, enforcing human alignment.
>
> Given the effective performance of the method proposed in this paper, it can be considered as a natural baseline for this RL-based extensions, meanwhile eliminating the need for time-consuming dataset management and extensive fine-tuning.
>
> **Comment 4**. Limitations. The authors have not sufficiently discussed potential negative societal impacts or broader ethical considerations associated with deploying such a system in real-world, high-stakes environments. Additionally, overreliance on automated dispatching systems without proper human oversight may lead to safety concerns or reduced job control for drivers in mobility-on-demand platforms.
>
> **Response**: Thanks for the reviewer's comments. We acknowledge that automated decision-making systems must be designed carefully to avoid unintended biases. While our current focus is on optimizing measurable metrics such as waiting time, detour time, and load balance, we agree that incorporating fairness or bias mitigation constraints is a valuable future direction.
>
> Our framework is extensible and can incorporate fairness-aware objectives or support post-hoc audits to identify and address potential disparate impacts. In contrast to fully autonomous systems, our method follows a co-optimization paradigm: the LLM generates high-level objectives, and a solver ensures feasibility and handles execution. This design facilitates human-in-the-loop control and allows domain-specific constraints to be directly embedded in the optimization model.
>
> We believe early limitations should not hinder forward-looking research. Agentic AI has demonstrated significant progress in many domains, and holds strong potential for intelligent transportation. Our system’s ability to dynamically learn and adapt dispatch strategies offers clear social benefits, reducing passenger wait times (**~16% improvement**), boosting efficiency, and easing city-wide congestion.

---

### Note · Authors · 2025-08-11

We thank the reviewers and AC for their thoughtful feedback and careful reading of our paper, **Hierarchical Optimization via LLM-Guided Objective Evolution for Mobility-on-Demand Systems**.

Our work proposes **the first closed-loop integration** of **LLM** with **mathematical optimization** for **sequential, dynamic decision-making**. The **LLM** serves as a **meta-optimizer**, generating high-level, interpretable objectives via prompt reasoning, while a **optimization solver** enforces **feasibility and numerical rigor**. We further introduce **a novel harmony search–based objective refinement mechanism** that iteratively evolves LLM-generated objectives using optimizer feedback. This **training-free framework** achieves state-of-the-art performance across diverse urban mobility scenarios, as validated on large-scale **New York** and **Chicago** taxi datasets. Extensive experiments (including ablations, sensitivity studies, and new large-scale tests with up to **1000 taxis**) show **∼16%** average reductions in passenger wait times and strong robustness to spatial and temporal heterogeneity.

We sincerely appreciate the reviewers’ detailed and constructive feedback. We are grateful that they recognized our novel integration of LLMs and optimization, training-free practicality, clarity, and comprehensive real-world evaluation. For example, R-E53E praised our “**creative integration of NLP and optimization methods**” and “**comprehensive evaluation**,” R-7QZG noted the “**novel and practical training-free scheme**” with consistent SOTA improvements, and R-ap7C affirmed the “**comprehensive and promising**” empirical results. R-8CYJ valued our novel problem formulation: **first integration of LLMs with exact solvers** for sequential dynamic decision-making.

We have provided detailed responses to all reviewer questions, including: plans to incorporate richer dynamic factors (traffic congestion, road disruptions); deeper comparisons with SOTAs; large-scale scalability experiments; runtime feasibility analysis; clarification of harmony search design; and ethical considerations, fairness constraints, and human-in-the-loop integration. All concerns have been addressed with additional experiments, runtime statistics, revised figures, and expanded methodological justifications.

We believe these clarifications and results further strengthen the novelty, practicality, and robustness of our proposed framework.

---

### Decision · Program_Chairs · 2025-09-17

**Decision:**

Accept (poster)

**Comment:**

This paper proposes a hierarchical optimization framework, and four reviewers have submitted their recommendations for the manuscript. This paper it the first closed-loop integration of LLM with mathematical optimization for sequential, dynamic decision-making. Although everyone has pointed out some merits, they also have raised some concerns, such as the motivation for HS algorithms, scalability and its application to real-world scenarios. During the rebuttal period, the authors’ feedback has helped on clarifying the reviewers’ concerns. Thus, the average score 4 is both above the average levels.